# Phase 1 dose expansion and biomarker study assessing first-in-class tumor microenvironment modulator VT1021 in patients with advanced solid tumors

## Abstract

**Background** Preclinical studies have demonstrated that VT1021, a first-in-class therapeutic agent, inhibits tumor growth via stimulation of thrombospondin-1 (TSP-1) and reprograms the tumor microenvironment. We recently reported data from the dose escalation part of a phase I study of VT1021 in solid tumors. Here, we report findings from the dose expansion phase of the same study.

**Methods** We analyzed the safety and tolerability, clinical response, and biomarker profile of VT1021 in the expansion portion of the phase I study (NCT03364400). Safety/tolerability is determined by adverse events related to the treatment. Clinical response is determined by RECIST v1.1 and iRECIST. Biomarkers are measured by multiplexed ion beam imaging and enzyme-linked immunoassay (ELISA).

**Results** First, we report the safety and tolerability data as the primary outcome of this study. Adverse events (AE) suspected to be related to the study treatment (RTEAEs) are mostly grade 1–2. There are no grade 4 or 5 adverse events. VT1021 is safe and well tolerated in patients with solid tumors in this study. We report clinical responses as a secondary efficacy outcome. VT1021 demonstrates promising single-agent clinical activity in recurrent GBM (rGBM) in this study. Among 22 patients with rGBM, the overall disease control rate (DCR) is 45% (95% confidence interval, 0.24-0.67). Finally, we report the exploratory outcomes of this study. We show the clinical confirmation of TSP-1 induction and TME remodeling by VT1021. Our biomarker analysis identifies several plasmatic cytokines as potential biomarkers for future clinical studies.

**Conclusions** VT1021 is safe and well-tolerated in patients with solid tumors in a phase I expansion study. VT1021 has advanced to a phase II/III clinical study in glioblastoma (NCT03970447).

## Plain language summary

The network of cells that surround a tumor, the tumor microenvironment, can help cancers to grow. Therapies targeting the tumor microenvironment may help to stop tumor growth. One such therapy is VT1021. In animal models, VT1021 treatment stops tumor cells from growing by changing the tumor microenvironment. Here, we have tested VT1021 in a clinical trial and found that VT1021 treatment is safe and well tolerated in patients with cancer. We also see signs of efficacy in some patients and observe evidence that VT1021 modifies the tumor microenvironment, which may help to block tumor growth. Finally, we identified several markers from the blood that may help to predict which patients will best benefit from VT1021 treatment. With further testing in clinical trials, VT1021 may be a useful therapy for patients with cancer.

VT1021, a cyclic 5-amino acid peptide designed to stimulate thrombospondin-1 (TSP-1) expression by replicating the biological activity of prosaposin, has been shown to stimulate TSP-1 production in the tumor microenvironment (TME)[1]. TSP-1 is a large glycoprotein predominantly localized in the extracellular matrix (ECM). Its biological functions include regulations of inflammation, immune response, wound healing, angiogenesis, and carcinogenesis[2–6]. With regard to TSP-1's role in tumor inhibition, it has been reported to directly inhibit tumor growth by inducing apoptosis in tumor cells or indirectly via reprograming the TME[7]. TSP-1 has also been shown to directly kill tumor cells via binding to CD36, resulting in apoptosis, as well as enhancing macrophage killing of tumor cells via binding to CD47[8–10]. TSP-1 also remodels the TME via several mechanisms including reprogramming macrophages, stimulating the infiltration of T cells and inhibiting angiogenesis[7,11]. Among these mechanisms, the role of TSP-1 in angiogenesis has been intensively studied and has previously been shown to be a potent endogenous inhibitor of angiogenesis in the TME[4,5] via

✉ e-mail: jenny.chen@vigeotx.com; jing.watnick@vigeotx.com

its interaction with two cell surface receptors, CD47 and CD36, as well as by antagonizing VEGF signaling[12–14].

TSP-1 levels are downregulated in a variety of cancers and low TSP-1 levels in the TME have been shown to positively correlate with growth and metastasis in several different cancer types[15–19]. Furthermore, TSP-1 expression in the TME is repressed in metastatic tumors compared to localized primary tumors[1]. It has also been shown that TSP-1 induction inhibits angiogenesis and induces apoptosis[13,14,20,21]. Thus, TSP-1 induction in the TME has been proposed as a potential therapeutic strategy to target tumor angiogenesis, growth, and metastasis[6]. In preclinical studies, VT1021 inhibited tumor growth via TSP-1 stimulation that resulted in the reprogramming of the TME from immunosuppressive and tumor-promoting to immune active and tumor-inhibiting[1,10,22]. Additionally, VT1021 has been tested in a phase I clinical study in solid tumors (NCT03364400)[23–26].

Here, we provide the safety profile and clinical outcome of VT1021 in a phase I expansion study in patients with solid tumors (NCT03364400). To confirm the mechanism of action (MOA) of VT1021 in the clinical setting and to identify the patient population that benefits from VT1021 treatment, we have analyzed potential biomarkers from circulating blood samples and available paired biopsy samples from evaluable patients enrolled in the phase I expansion study. Here, we report the clinical confirmation of TSP-1 induction by VT1021 in peripheral blood mononuclear cells (PBMCs) and the TME. Moreover, using multiplex ion beam imaging (MIBI) of paired tumor biopsy samples, we have observed modifications in the TME that correlated with VT1021 treatment including increased ratio of $CD8^+$ T cells (cytotoxic T lymphocytes, CTL) to T Regulatory (Treg) cells, increased $CD8^+$ Tumor-infiltrating lymphocytes (TIL), decreased T cell exhaustion, increased M1:M2 macrophage ratio and decreased microvascular density. Our data supports the role of TSP-1 in reprogramming the TME to inhibit tumor growth. To identify robust non-invasive biomarkers for VT1021, we have analyzed plasmatic cytokines and identified four cytokines, matrix metallopeptidase 9 (MMP9), plasminogen activator inhibitor-1 (PAI-1), C-C Motif Chemokine Ligand 5 (CCL5), and chitinase 3 like protein 1 (CHI3L1) whose baseline levels correlated with clinical responses, suggesting they are potential predictive biomarkers for VT1021. In addition, we have identified five potential pharmacodynamic biomarkers, including PAI-1, macrophage migration inhibitory factor (MIF), Interleukin-18 binding protein alpha (IL-18 Bpa), CCL5 and CHI3L1. Taken together, we provide the clinical outcome of VT1021 in a phase I expansion study in patients with solid tumors and report the clinical confirmation of TSP-1 induction by VT1021. Importantly, the TME modifications observed in the study support the role of VT1021 in reprogramming the TME from immunosuppressive to immune active. Lastly, several plasmatic cytokines have been identified as potential predictive and pharmacodynamic biomarkers for VT1021.

## Methods

### Clinical study design
The clinical trial was approved by the site's institutional review board and conducted in accordance with the principles of Good Clinical Practice and federal guidelines for human investigations. The IRBs that approved the study and their approval numbers are as follow: Horizon Oncology (420170357), START (CR00298293), Northwestern University (STU00207309-MOD0038), Sarah Cannon (RM 646), John Hopkins (IRB00208082), Mass General Brigham and Dana-Farber (18-609), Cleveland Clinic/Case Western University Hospitals (STUDY20191429), and MD Anderson (2019-1149_MOD001). All participants provided their written informed consent before enrollment. The study was registered on clinicaltrials.gov (NCT03364400).

In the expansion phase, patients were enrolled into one of the five cohorts: recurrent glioblastoma (rGBM), pancreatic cancer, ovarian cancer, TNBC and a high-CD36 and high-CD47 basket cohort. Key inclusion and exclusion criteria have been published previously (NCT03364400). All patients received VT1021 (11.8 mg/kg, twice a week intravenously) until disease progression. Clinical response was evaluated by using RECIST v1.1 and iRECIST in a population of patients who have at least completed one cycle of VT1021 treatment. Response assessment by CT or MRI was performed every 8 weeks until it was determined that the patient had progressive disease (PD). Tumor response was evaluated every 8 weeks ± 1 week according to RECIST guidelines (version 1.1) and iRECIST guidelines, or RANO guidelines for rGBM patients with iRANO modifications. Clinical benefit was defined by best overall response with complete response (CR), partial response (PR), or stable disease (SD) > 2 cycles (8 weeks). For the dose expansion study, the first patient was enrolled on March 24, 2020, and the last patient was enrolled on March 1, 2021. The primary objective of the expansion phase was to characterize the safety and tolerability of VT1021 in patient cohorts of specific indications. Secondary objectives were to characterize the adverse event (AE) profile; determine the pharmacokinetics of VT1021; describe preliminary evidence of efficacy using objective response rate (ORR), disease control rate (DCR), and progression-free survival (PFS) based on RECIST v1.1 or Response Assessment in Neuro-Oncology (RANO) with iRANO modifications for GBM patients; and determine overall response rate by iRECIST. Exploratory Objectives were to determine the pharmacodynamics (PD) of VT1021 including the effect of VT1021 on TSP-1 and TME. For clinical response and biomarker studies reported here, a patient ID number is assigned to each patient.

### Inclusion criteria
To qualify for enrollment, all the following criteria must be met: (1) Patient must provide written informed consent. (2) Patient is ≥18 years of age. (3) For the Dose Expansion Phase: Patients with advanced solid tumors that are refractory to existing therapies known to provide clinical benefit for their condition. Also, a patient may be intolerant of, not eligible for, or has refused prior standard of care therapies. (4) Patient has evaluable or measurable disease by RECIST v1.1. or, for patients with GBM, RANO. (5) Patient has a performance status (PS) of 0-1 on the Eastern Cooperative Oncology Group (ECOG) scale or in the case of GBM patients Karnofsky PS of ≥60%. (6) Patient is at least 21 days removed from therapeutic radiation or chemotherapy prior to the first scheduled day of dosing with VT1021 and has recovered to Grade ≤ 1 (National Cancer Institute [NCI] Common Terminology Criteria for Adverse Events [CTCAE] v5.0) from all clinically significant toxicities related to prior therapies. (a) For patients receiving nitrosoureas or mitomycin C, the window is 6 weeks. (b) For patients receiving monoclonal antibody therapy, the window is at least one half-life or 4 weeks (whichever is shorter). (7) Patient has adequate organ function defined as: (a) Absolute neutrophil count (ANC) ≥ 1.5 × 109/L (1500/μL) and absolute lymphocyte count (ALC) ≥ 7 × 109/L (700/μL). (b) Platelet ≥ 100 × 109/L. (c) Hemoglobin ≥ 9 g/dL. (d) Activated partial thromboplastin time/ prothrombin time/international normalized ratio (aPTT/PT/INR) ≤ 1.5 × upper limit of normal (ULN) unless the patient is on anticoagulants in which case therapeutically acceptable values (as determined by the investigator) meet eligibility requirements. (e) Aspartate aminotransferase (AST) or alanine aminotransferase (ALT) ≤ 2.5 × ULN. In the case of known (i.e., radiological or biopsy documented) liver metastasis, serum transaminase levels must be ≤5 × ULN. (f) Total serum bilirubin ≤1.5 × ULN (except for patients with known Gilbert's Syndrome ≤3 × ULN is permitted). (g) Renal: Serum creatinine < 2.0 × ULN and creatinine clearance ≥50 mL/min/1.73 m². (h) Serum albumin > 3 gm/dL. (8) Patient agrees to use acceptable methods of contraception during the study and for at least 90 days after the last dose of VT1021 if sexually active and able to bear or beget children.

### Exclusion criteria
The presence of any of the following will exclude the patient from the study: (1) Diagnosis of another malignancy within the past 2 years (excluding a history of carcinoma in situ of the cervix, superficial non-melanoma skin cancer, or superficial bladder cancer that has been adequately treated, or stage 1 prostate cancer that does not require treatment or requires only treatment with luteinizing hormone-releasing hormone agonizts or antagonists if initiated at least 90 days prior to the first dose of VT1021). (2)

History of a major surgical procedure or a significant traumatic injury within 14 days prior to commencing study drug, or the anticipation of the need for a major surgical procedure during the course of the study. (3) Treatment with investigational therapy(ies) within 5 half-lives of the investigational therapy prior to the first scheduled day of dosing with VT1021, or 4 weeks if the half-life of the investigational agent is not known, whichever is shorter. (4) Concurrent serious (as determined by the Principal Investigator [PI]) medical conditions, including, but not limited to, New York Heart Association (NYHA) class III or IV congestive heart failure, history of congenital prolonged QT syndrome, uncontrolled infection, active hepatitis B, hepatitis C or human immunodeficiency virus (HIV), or other significant co-morbid conditions that, in the opinion of the Investigator, would impair study participation or cooperation. (5) Pregnant or planning to become pregnant or breast feed while on study. (6) Evidence of symptomatic brain metastases. Patients with treated (surgically excised or irradiated) and stable brain metastases are eligible, assuming the patient has adequately recovered from treatment, the treatment was at least 28 days prior to initiation of study drug, and baseline brain computed tomography (CT) with contrast or magnetic resonance imaging (MRI) within 14 days of initiation of study drug, is negative for new or worsening brain metastases. (7) Other concurrent chemotherapy, immunotherapy, radiotherapy, or investigational anti-cancer therapy. (8) Requirement for palliative radiotherapy to lesions that are defined as target lesions by RECIST/RANO criteria at the time of study entry. (9) Known hypersensitivity to any of the components of VT1021 (sodium phosphate dibasic anhydrous, sodium phosphate monobasic monohydrate, mannitol, polysorbate 80) or a severe reaction to PS20- or PS80-containing drugs or investigational agents (e.g., amiodarone, Vitamin K, etoposide, docetaxel, cancer vaccine, protein biotherapeutics [like monoclonal antibodies], erythropoietin-stimulating agents, fosaprepitant). (10) Chronic, systemically administered glucocorticoids in doses equivalent to >5 mg prednisone daily. Topical, inhalational, ophthalmic, intraarticular, and intranasal glucocorticoids are permitted. Isolated or intermittent use of systemically administered glucocorticoids to treat complications of malignancy, use as a premedication, or as a one-time prep for an imaging procedure is permitted. If patient was on >5 mg prednisone/day equivalent, last dose must have been at least 7 days prior to the first planned dose of study drug. Exception: GBM patients may be on chronically administered glucocorticoids for the control of cerebral edema as long as the dose does not exceed 2 mg dexamethasone/15 mg prednisone/day. For GBM patients requiring larger amounts of glucocorticosteroids, consultation with and agreement by the medical monitor is required before such patients can be enrolled. (11) Patients with active hepatitis B (e.g., hepatitis B surface antigen [HBsAg] reactive) are excluded, however, patients with past hepatitis B virus (HBV) infection or resolved HBV infection (defined as the presence of hepatitis B core antibody [HBcAb] and absence of HBsAg) may be enrolled provided that prior testing/known status for HBV deoxyribonucleic acid (DNA) is negative. Patients with active hepatitis C (e.g., hepatitis C virus [HCV] ribonucleic acid [RNA] [qualitative] are detected) are excluded, however, patients with cured hepatitis C (negative HCV RNA prior test/known status) may be enrolled.

### Clinical sample acquisition

Blood samples for ELISA and quantitative RT-PCR (qRT-PCR) assays were collected from each patient in the expansion phase at 15 timepoints: pre, 0 h, 2 h, 4 h, and 6 h on days 1, 4, and 53 post-dosing with VT1021. Biomarkers analyzed from clinical samples collected on day 1 pre-dose (pre) timepoint are referred to as the baseline levels. For ELISA assays, SepMate-based fractionation was performed to isolate PBMCs, platelets and plasma from whole blood using SepMate-50 tubes (Stemcell Technologies, cat no. 15415) by a CRO according to the manufacturer's instructions. This isolation protocol has been previously used by others[27,28]. For qRT-PCR assays, blood samples were collected in PAXgene Blood RNA Tubes (BD, cat. no. 762165) according to the manufacturer's instructions. This method has been used to analyze RNA from blood samples for biomarker studies[29]. Blood samples from all evaluable patients ($n = 46$) were used for analysis; note that sample

deviations (30%) occurred due to the COVID-19 pandemic (2019–2021), some samples were not collected at clinical sites as scheduled or some samples were damaged during the shipping process.

For patients who signed a consent form, either a pre-treatment biopsy or archival tumor specimen are considered the Screen biopsy. In addition, in-treatment (In-Tx) biopsies were collected at the end of Cycle 1 Week 4, or at any time during Cycle 2, or thereafter at the discretion of the Investigator.

### ELISA assays for TSP-1 and plasmatic cytokines

For the TSP-1 ELISA assay, TSP-1 protein levels in PBMCs, platelets and plasma were assessed by ELISA using Human Thrombospondin-1 Quantikine ELISA Kit (R&D, cat# DTSP10) according to the manufacturer's instruction. To analyze other plasmatic cytokines, Human XL Cytokine Array Kit (R&D, cat# ART022B) was used for a primary screen according to the manufacturer's instruction followed by ELISA assays to confirm the results. ELISA kits were purchased and used according to the manufacturer's instructions: MMP9 (R&D, cat# DMP900), PAI-1 (R&D, cat# DSTE100), CHI3L1 (R&D, cat# DC3L10), CCL5 (Invitrogen, cat# EHRNTS), MIF (R&D, cat# DMF00B), IL18 Bpa (R&D, cat# DBP180). Due to the variation of input blood volume collected for each sample, the total amount of analytes per blood sample was calculated and normalized by input blood volume. For each patient, baseline and maximum or minimum level of the analytes were recorded and used for the statistical analysis (see Supplementary Fig. 1 for examples of cytokine changes over time). There are several variables and technical issues associated with collection and analysis of the blood samples in this clinical study. The two most commonly encountered issues also happened to be outside the ability to control. Specifically, due to the COVID-19 pandemic, unpredictable delays in shipping, which negatively affected the quality of the samples; and difficulty with sample collection, which resulted in non-collection of samples at various time points. These issues created challenges for data analysis. Accordingly, we decided to conduct our analysis of cytokine levels based on maxima or minima compared to baseline. This analysis provides the clearest demonstration of the relevant biological activity of VT1021 and mitigates against potential outliers and unintentional bias.

### RNA extraction and quantitative RT-PCR assay

TSP-1 mRNA levels in peripheral blood mononuclear cells (PBMCs) were analyzed by quantitative RT-PCR following extraction of total RNA from patients' whole blood samples collected in PAXgene Blood RNA kit (Qiagen, cat # 762164). Briefly, RNA was extracted using PAXgene Blood RNA Kit according to the manufacturer's instruction. For cDNA synthesis, RNA to cDNA EcoDry Premix (Takara, cat# 639548) was used. cDNA synthesis was performed by MiniAmp Thermal Cycler detection system (Applied Biosystems). PCR was conducted on the QuantStudio 6 Pro (Applied Biosystems) according to the manufacturer's protocol. The TaqMan gene expression assay system (Applied Biosystems) was used for quantifying the levels of TSP-1 and GAPDH. The Sequence Detection system QuantStudio6 Pro (Applied Biosystems) was used to analyze amplification plots. The relative quantity of amplified cDNA was calculated by interpolating an average of four replicate Ct values onto a standard curve of Ct values obtained from serially diluted cDNA and normalized for expression of GAPDH. Four independent qPCR results were analyzed for each sample, standard deviations were calculated.

### Multiplexed ion beam imaging of biopsy samples

Multiplexed ion beam imaging (MIBI) of paired pre-treatment (Screen) and in-treatment (In-Tx) biopsies was performed by Ionpath (CRO). Briefly, gold slides with 5-micron tissue sections were placed in a slide rack and baked at 70 °C for 20 min. Slides were rehydrated and incubated in 1× HIER buffer in the PT Module for heat-induced epitope retrieval (HIER). Slides were rinsed in 1× TBST and a solid border around the tissue section was drawn with a PAP pen. Slides were rinsed again with 1× TBST using the PAP pen border, and tissue was incubated with 100–200 μL blocking buffer for 20 min at room temperature in a moisture chamber. Next, 100–200 μL of

the metal-conjugated antibody panel per tissue area was prepared. The blocking buffer was removed and 100–200 µL of antibody master mix was added to each tissue section. Slides were incubated in the moisture chamber overnight at 4 °C. The following day, the secondary antibody panel was prepared, and the primary antibody panel was removed and washed in 1× TBST. 100-200 µL of the secondary antibody panel was added to each tissue section and incubated in the moisture chamber at 4 °C for 1 h. The secondary antibody was removed, the slides were washed with 1× TBST and dehydrated. Slides were dried in a desiccator for at least 1 h prior to MIBI scope analysis. Three 800 µm$^2$ regions of interest (ROIs) were selected for each biopsy sample for MIBI analysis.

Antibodies and their clones that were used for the MIBI are as follows: dsDNA (35I9 DNA), iNOS (SP126), CD163 (10D6), FOXP3 (236 A/E7), PD-1 (D4W2J), CD31 (EP3095), CD8 (C8/144B), CD3 (MRQ-39), TIM-3 (EPR22241), Keratin (AE1/AE3), TSP-1 (EPR22927-54). Ionpath (CRO) performed all the staining. For any technical details, please refer to https://www.ionpath.com.

## Statistical analysis

The disease control rate (DCR) used for clinical outcomes was calculated as the percentage of patients with advanced cancer whose therapeutic intervention has led to a complete response, partial response, or stable disease. The 95% confidence interval of DCR was calculated using Clopper-Pearson Exact method. The one-way analysis of variance (ANOVA) was used to analyze statistical significance in more than two groups. All bar graphs and scatter plots with bar (individual dot representing a patient) show the mean and standard error of mean (SEM) and were generated with GraphPad Prism 9.3.1. The p-value was calculated with Graphpad Prism 9.3.1 by paired or unpaired t-test, two tail. The significance level for the statistical tests was set at 5%. The 95% confidence Interval (CI) was calculated using the exact interval. The formula used for Confidence interval calculation: $(CI) = \bar{X} \pm Z(S \div \sqrt{n})$. $\bar{X}$ represents the sample mean, $Z$ represents the Z-value from the normal standard distribution, $S$ is the population standard deviation and $n$ represents the sample size.

## Results

### Demographics and characteristics

Preclinical studies have demonstrated that VT1021 potently inhibits tumor growth via stimulation of TSP-1 expression[1,10,22]. VT1021 has been tested in a phase I clinical study in solid tumors (NCT03364400). The phase 1 escalation study has been reported previously[26]. Here we report the clinical and biomarker analyses of a phase 1 expansion study. A total of 47 patients enrolled for this phase 1 expansion study were evaluable patients. The baseline demographics of the enrolled patients are presented in Table 1. The median age was 65 years with a range from 24 to 83 years. 21 (44.7%) of the 47 patients were male, while 26 (55.3%) patients were female. Patients enrolled in this study were required to have a performance status (PS) of 0-1 on the Eastern Cooperative Oncology Group (ECOG) scale or in the case of GBM patients Karnofsky PS (KPS) of ≥60%. There were 2 patients that had ECOG of 2, however both patients had a KPS of 70 which made them eligible. One patient did not have an ECOG but did have KPS of 70 making that patient eligible. The primary tumor types were recurrent glioblastoma (22, 46.8%) and metastatic pancreatic cancer (17, 36.7%). All other cancer types had a total of 8 patients including ovarian cancer (4, 8.5%), colorectal cancer (2, 4.3%), mesothelioma (1, 2.1%), and cholangiocarcinoma (1, 2.1%). All patients enrolled for this study were at metastatic stage.

### Safety and tolerability

Adverse events (AE) that were suspected to be related to the study treatment (RTEAEs) were mostly grade 1–2 (Table 2). There were no grade 4 or 5 adverse events. RTEAEs were experienced by 29 patients (61.74%) shown in Table 2; the most frequent RTEAEs (>10% of patients) were diarrhea (9 patients, 19.1%), fatigue (7 patients, 14.9%), and nausea (6 patients, 12.8%). Grade 3 RTEAEs were reported in 3 patients (6.4%): patient #28 (diarrhea,

**Table 1 | Patient demographics and baseline characteristics**

| Characteristics | $n = 47$ |
|---|---|
| Median age, years | 65 (24–83) |
| *Gender n (%)* | |
| Male | 21 (44.7%) |
| Female | 26 (55.3%) |
| *Eastern Oncology Group Performance Status, n (%)* | |
| 0 | 11 (23.4%) |
| 1 | 33 (70.2%) |
| 2* | 2 (4.3%) |
| None** | 1 (2.1%) |
| *Primary tumor type, n (%)* | |
| Glioblastoma | 22 (46.8%) |
| Pancreatic cancer | 17 (36.7%) |
| Ovarian cancer | 4 (8.5%) |
| Colorectal cancer | 2 (4/3%) |
| Mesothelioma | 1 (2.1%) |
| Cholangiocarcinoma | 1 (2.1%) |
| Metastasis, *n* (%) | 47 (100%) |

*There were 2 patients that had an ECOG of 2, however both patients had a KPS of 70 which made them eligible. **One patient did not have an ECOG but did have KPS of 70 making that patient eligible.

ovarian cancer), patient #43 (nausea, pancreatic cancer), and patient #2 (lipase increased, GBM), and these patients were on study for 63, 49, and 530 days, respectively (Fig. 1b). In summary, VT1021 was safe and well tolerated in the expansion study, which is consistent with the clean safety profile observed in the escalation study[26].

### Clinical response

The phase I study design for VT1021 is shown in Fig. 1a. The data set of the escalation phase has been reported previously[26]. Briefly, VT1021 was administrated at 0.5–15.6 mg/kg twice weekly intravenous (IV) infusions in patients with solid tumors. The RP2D has been determined as 11.8 mg/kg. For the expansion phase, VT1021 was administrated at 11.8 mg/kg twice weekly IV infusions in patients with rGBM ($n = 22$), pancreatic cancer ($n = 17$), and other tumor types ($n = 8$). The clinical responses of the phase I expansion study for 47 evaluable patients are depicted using a swimmer plot in Fig. 1b. For each patient, the primary tumor type and patient ID are shown and the duration of the treatment (days on treatment) is represented by a bar. The best clinical responses are color coded (Fig. 1b). Among 22 evaluable patients with rGBM, 3 had complete response (CR), 1 had partial response (PR), and 6 had stable disease (SD) with an average study duration of over 120 days (Fig. 1b). One patient (GBM, #1) has been on VT1021 treatment for >1100 days with no measurable lesion detectable. This patient is currently under single patient IND (Investigational New Drug) (Fig. 1b). The overall disease control rate (DCR) for rGBM was 45% (95% CI, 0.24–0.67). Among the 17 evaluable pancreatic cancer patients, 3 achieved SD with a DCR of 18% (95% CI, 0.04–0.43). Among 8 patients with other cancer types, 2 patients with ovarian cancer achieved SD. Therefore, since a total of 15 patients out of all 47 patients achieved CR/PR/SD, the DCR was 32% (95% CI, 0.19–0.47) for the expansion phase (Supplementary Data 1). Based on the well-established safety profile and beneficial clinical outcome of VT1021 in rGBM, VT1021 has advanced to a phase II/III clinical study in glioblastoma (GBM AGILE; NCT03970447).

### TSP-1 was induced in evaluable patients after VT1021 treatment

To investigate whether TSP-1 is induced by VT1021 in the clinical setting, TSP-1 levels were analyzed in blood samples from all evaluable patients enrolled in the phase I expansion study[24,25]. One patient (#26) had day

## Table 2 | Related treatment-emergent adverse events

| AEs, n (%) | Evaluable patient (n = 47) | |
|---|---|---|
| | All | Grade ≥ 3 |
| Total (patients with at least one event) | 29 (61.7) | 3 (6.4) |
| Diarrhea | 9 (19.1) | 1 (2.1) |
| Fatigue | 7 (14.9) | 0 |
| Nausea | 6 (12.8) | 1 (2.1) |
| Infusion related reaction | 4 (8.5) | 0 |
| Abdominal pain | 3 (6.4) | 0 |
| Decreased appetite | 3 (6.4) | 0 |
| Anemia | 2 (4.3) | 0 |
| Flatulence | 2 (4.3) | 0 |
| Vomiting | 2 (4.3) | 0 |
| Alanine aminotransferase increased | 2 (4.3) | 0 |
| Aspartate aminotransferase increased | 2 (4.3) | 0 |
| Blood alkaline phosphatase increased | 2 (4.3) | 0 |
| Lipase increased | 2 (4.3) | 1 (2.1) |
| Neutrophil count decreased | 2 (4.3) | 0 |
| Hyperuricaemia | 2 (4.3) | 0 |
| Arthralgia | 2 (4.3) | 0 |
| PAIN: CHEST | 1 (2.1) | 0 |
| Vision blurred | 1 (2.1) | 0 |
| Anal incontinence | 1 (2.1) | 0 |
| Colitis | 1 (2.1) | 0 |
| Constipation | 1 (2.1) | 0 |
| Dyspepsia | 1 (2.1) | 0 |
| Gastritis | 1 (2.1) | 0 |
| Salivary hypersecretion | 1 (2.1) | 0 |
| Non-cardiac chest pain | 1 (2.1) | 0 |
| Pyrexia | 1 (2.1) | 0 |
| Activated partial thromboplastin time prolonged | 1 (2.1) | 0 |
| Amylase increased | 1 (2.1) | 0 |
| Blood creatinine increased | 1 (2.1) | 0 |
| Weight decreased | 1 (2.1) | 0 |
| White blood cell count decreased | 1 (2.1) | 0 |
| Hyperkalaemia | 1 (2.1) | 0 |
| Hypoalbuminaemia | 1 (2.1) | 0 |
| Hypokalaemia | 1 (2.1) | 0 |
| Hypomagnesaemia | 1 (2.1) | 0 |
| Hyponatraemia | 1 (2.1) | 0 |
| Muscular weakness | 1 (2.1) | 0 |
| Dizziness | 1 (2.1) | 0 |
| Peripheral sensory neuropathy | 1 (2.1) | 0 |
| Confusional state | 1 (2.1) | 0 |
| Dysphonia | 1 (2.1) | 0 |
| Dyspnea | 1 (2.1) | 0 |
| Dyspnea exertional | 1 (2.1) | 0 |
| Pruritus | 1 (2.1) | 0 |
| Skin lesion | 1 (2.1) | 0 |
| Flushing | 1 (2.1) | 0 |
| Hypertension | 1 (2.1) | 0 |

Treatment-emergent adverse events (TEAE) is defined as any event that occurs on or after the first dose of study drug administration or any pre-existing event which worsened in severity after dosing.
Related includes all events other than those unlikely or unrelated.
Total = patients with at least one event.
Evaluable patient = patients who have completed first cycle of treatment.
Patients with multiple unique events are counted once per each unique preferred term.
Coding used MedDRA version 23.0

1 samples damaged, therefore the baseline biomarker data is only available for 46 patients.

We assessed cell-associated TSP-1 protein levels by ELISA in PBMCs and platelets, as well as secreted TSP-1 remaining in the plasma fraction after isolation of PBMCs, from patients with GBM (n = 22), pancreatic cancer (n = 16) and other tumor types (n = 8). Induction of TSP-1 by VT1021 was measured at the transcriptional level via RT-PCR measurement of whole blood samples (Fig. 2). A 2.8–4.5-fold TSP-1 induction at the mRNA level in PBMCs was observed in patients with GBM, pancreatic cancer and other tumor types (Fig. 2a). Induction of cell-associated and secreted TSP-1 by VT1021 was confirmed at the protein level via ELISA assay (Fig. 2b–d). We observed a 2.7–3.4-fold induction of cell-associated TSP-1 protein in PBMCs (Fig. 2b) along with a 1.6–6.8-fold upregulation of secreted TSP-1 protein (Fig. 2c). A 2.4–4.1-fold upregulation of TSP-1 protein was observed in platelet samples (Fig. 2d). Taken together, TSP-1 induction by VT1021 has been observed for all cancer types tested here, and no significant difference was observed between different cancer types (Fig. 2, Supplementary Data 2). Up-regulation of TSP-1 levels was observed in all evaluable patients following treatment with VT1021 across multiple indications in the phase I expansion study, confirming a role of VT1021 in TSP-1 induction in the clinical setting.

To understand the biological significance of TSP-1 induction by VT1021, TSP-1 levels were measured in pre- and in-treatment blood samples from evaluable patients and blood samples collected from healthy donors (Table 3). Of note, we found that cell-associated TSP-1 levels in PBMCs (0.80 μg/ml) and platelets (5.61 μg/ml) in healthy donors were significantly higher than baseline TSP-1 levels in PBMCs (0.29, 0.47 μg/ml) and platelets (1.18, 0.83 μg/ml) in evaluable patients with GBM and pancreatic cancer (Table 3). After VT1021 treatment, the mean TSP-1 protein levels in PBMCs (0.77, 0.76 μg/ml) and platelets (1.80, 1.24 μg/ml) were significantly upregulated in patients with GBM and pancreatic cancer (Table 3), suggesting that TSP-1 is a pharmacodynamic biomarker of VT1021 for patients with GBM and pancreatic cancer. Before VT1021 treatments, the baseline TSP-1 level in PBMCs from evaluable patients was 36%-66% of healthy donors; after VT1021 treatment, this ratio was 96–119% (Table 3). For baseline levels of secreted TSP-1, no significant difference was observed between healthy donors (4.53 μg/ml) and evaluable patients (4.30, 5.95, 4.51 μg/ml) (Table 3). After VT1021 treatments, the mean secreted TSP-1 protein level was significantly upregulated in patients with GBM (7.05 μg/ml); and increased in patients with pancreatic cancer (8.35 μg/ml) and other tumor types (7.55 μg/ml) (Table 3).

To evaluate the potential correlation between TSP-1 expression and clinical outcome, we analyzed absolute TSP-1 levels and TSP-1 fold-induction with respect to clinical responses in all evaluable patients with CR/PR/SD or PD (Table 4). Notably, the majority of TSP-1 in collected blood samples was secreted by PBMCs as opposed to cell-associated (Table 4). This observation is consistent with TSP-1 being a secreted protein. Among all the comparisons, secreted TSP-1 induction showed significant differences between the CR/PR/SD and PD (Table 4). Secreted TSP-1 induction was significantly higher in CR/PR/SD (5.41-fold) versus PD (1.48-fold) (p = 0.018) (Table 4). In addition, it was observed that baseline levels of secreted TSP-1 were much lower in CR/PR/SD (2.95 μg/ml) compared to in PD (5.50 μg/ml) (Table 4), suggesting that patients with lower baseline secreted TSP-1 levels may benefit from VT1021 treatment. These findings suggest that while VT1021 stimulates cell associated TSP-1 in the PBMCs, based on protein and mRNA levels, secreted TSP-1 levels of collected patient samples have potential correlation to the clinical response. It will be a specific focus to test this hypothesis with additional samples from future clinical studies.

### Elevated TSP-1 signal intensity in the TME observed in patients after VT1021 treatment

In addition to measuring the induction of TSP-1 in patient blood samples, we also have a strong interest in the expression of TSP-1 in the TME and the remodeling its induction induces. As such, we sought to examine patient

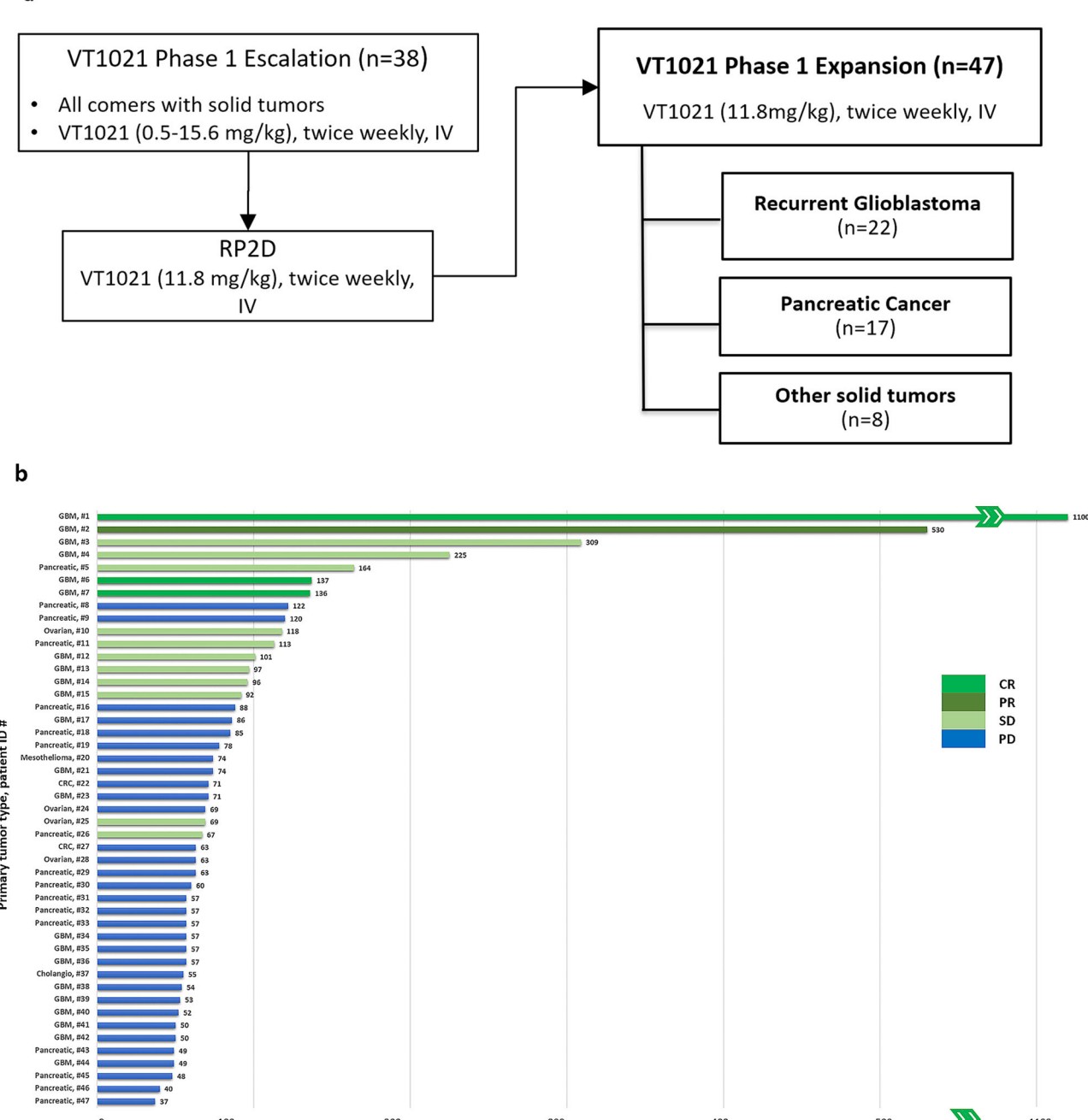

**Fig. 1 | The CONSORT diagram and clinical response for VT1021 in a phase I expansion study. a** The consort diagram for VT1021 in a phase I clinical trial. For the escalation study, 38 participants completed at least 2 cycles of VT1021 treatment. The RP2D of VT1021 has been determined as 11.8 mg/kg. The escalation study has been reported previously. For the expansion study, 47 participants completed at least 2 cycles of VT1021 treatment at 11.8 mg/kg, twice weekly, IV. **b** Swimmer plot showing best clinical responses of all evaluable patients (*n* = 47) enrolled in the phase 1 expansion study for VT1021. Primary tumor type and patient ID number are shown on the *y*-axis. The x-axis displays the days on treatment, the length of each bar represents the duration of the treatment of each patient. Bars are colored according to the best overall responses. CR complete response, PR partial response, SD stable disease, PD progressive disease, RP2D recommended phase II dose, GBM patients with glioblastoma, Pancreatic patients with pancreatic cancer, Ovarian patients with ovarian cancer, CRC patients with colorectal cancer. One patient (GBM, #1) with complete response has been on the study for over 1100 days and is currently under single patient IND (Investigational New Drug). "»" indicates the skipped labels from 600 to 1000 days on the *x*-axis.

tissue samples in a way that would allow us to visualize changes to the TME composition in a spatial and quantitative manner. Multiplex ion beam imaging (MIBI) is a powerful technique that allows the simultaneous analysis of multiple markers of interest from the same tissue section[30]. Antibodies conjugated with metal ions are incubated with biopsy sections on gold slides and detected by mass spectrometry to quantify their abundance in specified regions of interest (ROIs). Among 46 evaluable patients, only 9

paired biopsy samples were available and used for the MIBI analyses. The 9 paired biopsy samples included 2 SD patients (#10 and #11) and 7 PD patients (#24, #22 #9, #18, #31, #43 and #47) (Fig. 1). The TME image analysis is considered descriptive analysis due to the low number of paired biopsy samples.

To analyze TSP-1 expression in the TME, MIBI for TSP-1 was performed with paired biopsy samples from patients enrolled in the phase I

**Fig. 2 | Peripheral TSP-1 induction fold post-VT1021 in all evaluable patients in a phase I expansion study. a** TSP-1 mRNA induction fold in PBMCs. **b** TSP-1 protein induction fold in PBMCs. **c** TSP-1 induction fold in plasma samples. For other cancer types (*n* = 5), one datapoint (28.32) is an outlier and omitted from the graph. **d** TSP-1 induction fold in platelets. Fewer samples were analyzed due to sample deviation. *n* the number of patients available for the analysis, Mean average TSP-1 induction fold, PBMCs peripheral blood mononuclear cells, GBM patients with recurrent glioblastoma, Pancreatic patients with pancreatic cancer, Other patients with cancer types other than GBM or pancreatic cancer. Error bars indicate SEM.

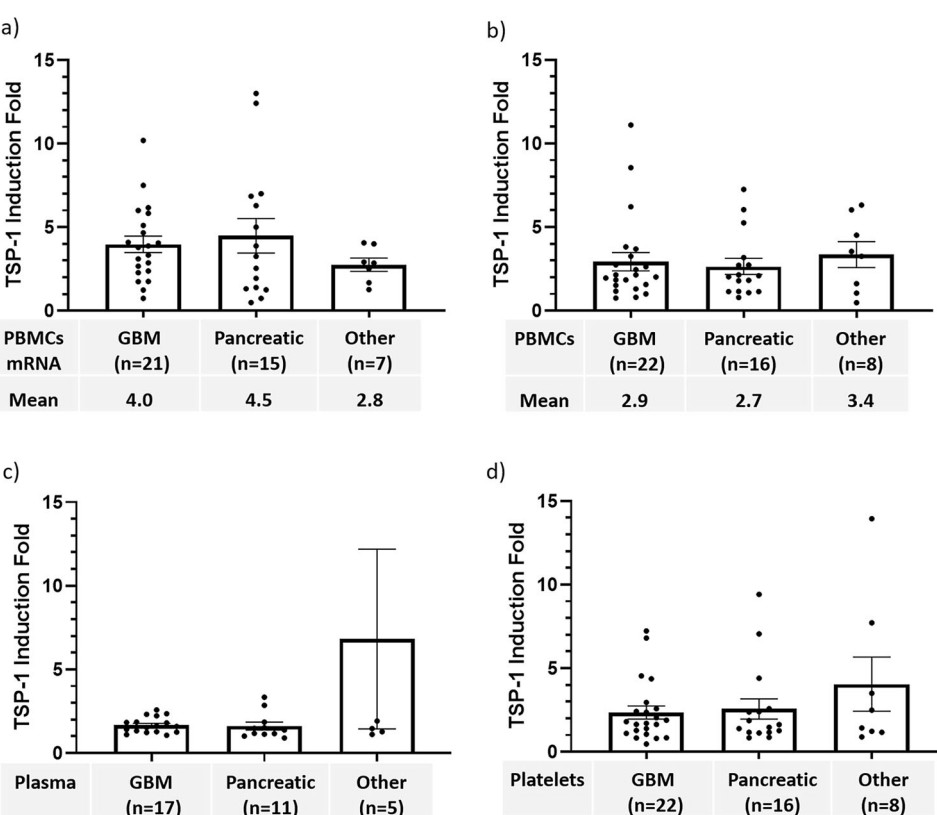

| PBMCs mRNA | GBM (n=21) | Pancreatic (n=15) | Other (n=7) |
|---|---|---|---|
| Mean | 4.0 | 4.5 | 2.8 |

| PBMCs | GBM (n=22) | Pancreatic (n=16) | Other (n=8) |
|---|---|---|---|
| Mean | 2.9 | 2.7 | 3.4 |

| Plasma | GBM (n=17) | Pancreatic (n=11) | Other (n=5) |
|---|---|---|---|
| Mean | 1.7 | 1.6 | 6.8 |

| Platelets | GBM (n=22) | Pancreatic (n=16) | Other (n=8) |
|---|---|---|---|
| Mean | 2.4 | 2.6 | 4.1 |

expansion study. TSP-1 signal density was quantified for paired biopsy samples from 9 patients (Supplementary Data 3). The average TSP-1 signal density was higher in the in-treatment (In-Tx) biopsy samples (324236/mm²) compared to the screen biopsy samples (199032/mm²) (Fig. 3a). The representative images from one patient (#24) are shown in Fig. 3b. These observations suggest that TSP-1 levels in the TME were increased after VT1021 treatment and are in agreement with the up-regulation of TSP-1 levels in peripheral blood samples following VT1021 treatment.

Preclinical studies with mouse models showed that VT1021 stimulated TSP-1 production in CD11b⁺ myeloid-derived suppressor cells (MDSCs) and contributed to its tumor-inhibitory function[10]. In agreement with those findings, we observed increased TSP-1 signal density in CD11b⁺ MDSCs in the in-treatment biopsy samples compared to screen biopsy samples in 4 (#11, #22, #24, #47) out of 9 patients analyzed with MIBI analysis; representative images from patient #24 are shown in Fig. 3b. We also noticed that TSP-1 upregulation in the TME after VT1021 treatment was not limited to MDSCs. Several other cell types, including T cells, macrophages, and tumor cells, also showed elevated TSP-1 expression in the TME after VT1021 treatment, which is consistent with previous reports that a variety of normal and malignant cells secrete TSP-1[2].

### Increased CTL:Treg ratio observed in the TME after VT1021 treatment

CTLs are major components of the TME and are capable of efficiently killing and eliminating cancer cells[31]. In contrast, Treg cells are a specialized subpopulation of T cells that suppress the immune response[32–34]. It has been proposed that an increased CTL:Treg ratio in the TME contributes to tumor growth inhibition[35]. To investigate whether VT1021 has an impact on TME programming, we analyzed the CTL:Treg ratio in the TME by MIBI. CTLs were detected using double positive CD3⁺ and CD8⁺ markers, and Tregs were detected using double positive CD3⁺ and FOXP3⁺ markers. We observed an increased CTL:Treg ratio in the TME in 5 (#10, #11, #22, #31,

#47) out of 9 patients after VT1021 treatment. Representative images for one patient (#10) are shown in Fig. 4b. In this patient, the CTL:Treg ratio was increased by 66% in the in-treatment biopsy sample compared to the screen biopsy sample as shown by quantification (Fig. 4a, Supplementary Data 4).

### Decreased CTL to tumor distance observed in the TME after VT1021 treatment

Tumor-infiltrating lymphocytes (TIL) play an essential role in mediating response to immuno-oncology (I-O) drugs and improving clinical outcomes in various cancer types[36,37]. To investigate whether VT1021 has an impact on TIL, we analyzed the CTL to tumor distance in the TME by MIBI. Tumor cells were detected using keratin⁺ markers. We observed a decreased CTL to tumor distance in the TME in 3 out of 9 patients (#10, #11, #43) after VT1021 treatment. Representative images for one patient (#10) are shown in Fig. 4e. In patient #10, the CTL to tumor distance was decreased by 60% in the in-treatment biopsy sample compared to the screen biopsy sample as shown by quantification (Fig. 4c, Supplementary Data 4). In this patient, CTLs were highly increased in the in-treatment biopsy sample compared to the screen biopsy sample as shown by quantification (Fig. 4d, Supplementary Data 4).

### Decreased exhausted T cell observed in the TME after VT1021 treatment

T cell exhaustion is one of the major mechanisms responsible for resistance or refractoriness to I-O drugs in the clinic[38,39]. Therefore, we investigated whether VT1021 had any impact on T cell exhaustion in the TME. Total exhausted T cells were detected by MIBI for CD3⁺ and PD-1⁺ T cells. Early exhausted T cells were detected by MIBI staining for CD3⁺, PD-1⁺, and TIM3⁻ T cells.

We observed decreased total exhausted T cell (CD3⁺PD-1⁺) and early exhausted T cell (CD3⁺PD-1⁺ TIM3⁻) density in the in-treatment biopsy samples compared to screen biopsy samples in 5 (#9, #11, #18, #31, #47) out

**Table 3 | Peripheral TSP-1 levels in healthy donors and evaluable patients enrolled in a phase I expansion study for VT1021**

| Peripheral blood samples | TSP-1 mean, (95% CI) | | TSP-1 ratio (%) (95% CI) | |
|---|---|---|---|---|
| | Baseline vs. max | | Patients vs. healthy donors | |
| | Baseline | Max | patients' baseline/healthy donors' baseline | Patients' max/healthy donors' baseline |
| *PBMCs* | | | | |
| Heathy donors (*n* = 51) | 0.80 (0.68–0.92) | | | |
| GBM (*n* = 22) | 0.29 (0.19–0.39) | 0.77 (0.42–1.12) | 36% (25–48%) | 96% (55–137%) |
| | | *p = 0.005 | *‡ p = 0.000001 | p = 0.83 |
| Pancreatic (*n* = 16) | 0.47 (0.16–0.77) | 0.76 (0.36–1.15) | 58% (23–93%) | 98% (54–144%) |
| | | *p = 0.010 | *‡ p = 0.016 | p = 0.79 |
| Others (*n* = 8) | 0.53 (0.04–1.02) | 0.95 (0.60–1.29) | 66% (15–116%) | 119% (83–154%) |
| | | p = 0.118 | p = 0.12 | p = 0.36 |
| *Platelets* | | | | |
| Heathy donors (*n* = 51) | 5.61 (4.96–6.25) | | | |
| GBM (*n* = 22) | 1.18 (0.81–1.55) | 1.80 (1.45–2.17) | 21% (15–27%) | 32% (26–38%) |
| | | *p = 0.0004 | *‡ p = <0.000001 | *‡ p = <0.000001 |
| Pancreatic (*n* = 16) | 0.83 (0.52–1.14) | 1.24 (0.97–1.52) | 15% (10–20%) | 23% (18–28%) |
| | | *p = 0.0006 | *‡ p = <0.000001 | *‡ p = <0.000001 |
| Others (*n* = 8) | 0.93 (0.18–1.67) | 1.55 (0.90–2.19) | 17% (6–28%) | 28% (18–37%) |
| | | p = 0.069 | *‡ p = <0.000001 | *‡ p = 0.00001 |
| *Plasma* | | | | |
| Heathy donors (*n* = 51) | 4.53 (3.79–5.27) | | | |
| GBM (*n* = 22) | 4.30 (3.44–5.16) | 7.05 (5.45–8.65) | 95% (77–112%) | 147% (115–181%) |
| | | *p < 0.0001 | p = 0.74 | *‡ p = 0.0020 |
| Pancreatic (*n* = 16) | 5.95 (3.37–8.53) | 8.35 (5.07–11.64) | 131% (81–181%) | 184% (121–248%) |
| | | p = 0.053 | p = 0.15 | *‡ p = 0.0006 |
| Others (*n* = 8) | 4.51 (−1.41 to 10.44) | 7.55 (−0.08 to 15.17) | 100% (7–192%) | 167% (48–285%) |
| | | p = 0.069 | p = 0.99 | *‡ p = 0.0411 |

*TSP-1 mean* the average TSP-1 level, *CI* confidence interval, *n* the number of patients available for the analysis. Fewer plasma samples were analyzed due to sample deviation.
Baseline: TSP-1 level analyzed from clinical samples collected on day 1 pre-dosing timepoint. Max: the maximum TSP-1 level from all timepoints excluding the day 1 pre-dosing timepoint.
ratio (%): the percentage of TSP-1 in patients (Baseline or Max) vs. TSP1-1 in healthy donors.
GBM: patients with recurrent glioblastoma. Pancreatic: patients with pancreatic cancer. Others: patients with cancer types other than GBM or pancreatic cancer.
*Significant difference observed in patients between the Baseline vs. Max; *p*-value calculated by paired *t*-test, two tail.
*‡: significant difference observed between patients vs. healthy donors; *p*-value calculated by unpaired *t*-test, two tail.

of 9 patients analyzed. The quantification of total and early exhausted T cell density, as well as the representative images for a representative patient are shown in Fig. 4. For patient #9, a ~75% reduction in total and early exhausted T cell density was observed in the in-treatment biopsy sample compared to the screen biopsy sample (Fig. 4f, g, Supplementary Data 4). The representative images for CD3, PD-1, and TIM3 are shown in Fig. 4h.

### Increased M1:M2 macrophage ratio observed in the TME after VT1021 treatment

Macrophages play pivotal roles in the innate immune response in the TME; M1-type macrophages promote immune response and inhibit tumor growth, whereas M2-type macrophages are immunosuppressive and promote tumor growth[40,41]. An increased M1:M2 macrophage ratio provides an immune promoting TME that inhibits tumor growth. Thus, we investigated whether VT1021 affected the M1:M2 macrophage ratio in the TME. MIBI was used to detect M1 macrophages by CD68[+] and iNOS[+], and M2 macrophages by CD68[+] and CD163[+].

From the MIBI analysis, we observed an increased M1:M2 macrophage ratio in the TME in 3 (#9, #22, #47) out of 9 patients after VT1021 treatment. The quantification of M1 and M2 macrophages, as well as the representative images for one patient (#22) are shown in Fig. 5. For patient #22, a 16-fold upregulation of the M1:M2 macrophage ratio was observed in the in-treatment biopsy sample compared to the screen biopsy sample (Fig. 5a,

Supplementary Data 5). The representative images by MIBI for CD68, iNOS, and CD163 are shown in Fig. 5b.

### Reduced microvascular density (MVD) observed in the TME after VT1021 treatment

TSP-1 is a potent endogenous inhibitor of angiogenesis in the TME[4,5]. It has been reported that TSP-1 expression negatively correlates with microvessel density (MVD)[14]. Therefore, we analyzed whether VT1021 had an impact on MVD, a surrogate marker for angiogenesis. Using MIBI, CD31 was used as a marker of vascular endothelial cells to indicate blood vessels. CD31[+] cells were counted to determine the MVD, regardless of the formation of the lumen.

We observed reduced CD31 density in 3 (#9, #18, #31) out of 9 patients after VT1021 treatment. The quantification of CD31 density, as well as the representative MIBI images for one patient (#18) are shown in Fig. 5. For patient #18, a reduction in CD31 density (35%) was observed in the in-treatment biopsy sample compared to the screen biopsy sample (Fig. 5c, Supplementary Data 5); representative images are shown in Fig. 5d.

In summary, we have analyzed the TME biomarkers in nine (2 SD and 7 PD) patients and observed TME modifications post-VT1021 in a subset of patients, and these changes support VT1021 in reprogramming the TME and inhibiting tumor growth. We were unable to correlate TME changes with clinical responses as biopsy pairs from CR/PR patients were unavailable.

## Plasmatic cytokines identified as potential biomarkers for VT1021

To identify robust non-invasive biomarkers for VT1021, we analyzed levels of specific plasmatic cytokines from patients with GBM and pancreatic cancer. We have screened a panel of 105 human cytokines using the plasma samples from 4 CR/PR patients and identified top hits with a fold change >1.5-fold after VT1021 treatment. We used the 1.5-fold as a cut-off because the plasmatic TSP-1 induction fold was >1.5-fold following VT1021 treatments (Table 5), and we intended to identify biomarkers with more robust changes than TSP-1 after VT1021 treatment. We identified several bio-

markers with higher fold changes than TSP-1, and then tested all top hits using plasma samples from rGBM ($n = 22$) and pancreatic cancer ($n = 17$) patients using ELISA kits. In summary, four potential predictive biomarkers, MMP9, PAI-1, CHI3L1 and CCL5, were identified from a panel of 105 human cytokines and further confirmed with ELISA assays (Table 5). Five potential pharmacodynamic biomarkers, including PAI-1, MIF, IL18 Bpa, CHI3L1 and CCL5 (Table 5) were identified from a panel of 105 human cytokines and further confirmed with ELISA assays (Table 5).

MMP9 is a pro-angiogenesis modulator that is negatively regulated by TSP-1[4,5]. The mean baseline MMP9 level in patients with CR/PR/SD (93.2 ng/ml) was significantly lower compared to PD patients (212.2 ng/ml), with a p-value of 0.01 by unpaired t-test, one tail (Table 5). Lower baseline MMP9 plasma levels correlated with better clinical response to VT1021 in patients with GBM and pancreatic cancer, suggesting that plasmatic MMP9 levels may be a predictive biomarker for VT1021. Consistent with our findings, data from a phase 3 study of Bevacizumab in newly diagnosed glioblastoma showed that baseline plasma MMP9 predicts overall survival (OS) benefit from Bevacizumab, where patients with lower MMP9 (<quartile 1) derived a significant 5.2-month OS benefit from Bevacizumab[42].

PAI-1 is a member of the Serpin superfamily of serine protease inhibitors involved in wound healing, fibrosis, angiogenesis, tumor cell invasion, and metastasis[43,44]. In our study, the mean baseline PAI-1 level in patients with CR/PR/SD was considerably lower compared to PD patients (8.5 ng/ml vs. 18.4 ng/ml) (Table 5). Lower baseline PAI-1 plasma levels correlated with better clinical response to VT1021, suggesting that plasmatic PAI-1 level may be a predictive biomarker for VT1021. A 1.5–2.6-fold reduction of PAI-1 was observed in all patients post-VT1021 treatment, suggesting that plasmatic PAI-1 could be a potential pharmacodynamic biomarker for VT1021 (Table 5). Our data is consistent with previous reports showing that high PAI-1 expression was correlated with poor prognosis in cancer patients[45].

CHI3L1, also known as YKL-40, is a secreted glycoprotein that plays a role in cancer cell growth, proliferation, invasion, metastasis, and angiogenesis[46]. Predictive and prognostic values were reported for YKL-40 in patients with rGBM and high-grade gliomas[47–50]. In our study, lower baseline CHI3L1 levels were observed in patients with CR/PR/SD (47.4 ng/ml) compared to PD patients (64.7 ng/ml), suggesting that plasmatic CHI3L1 level may be a predictive biomarker for VT1021 in patients with GBM and pancreatic cancer (Table 5). However, a 2.0-2.4-fold reduction of CHI3L1 was observed in all patients post-VT1021 treatment, regardless of clinical response, suggesting that plasmatic CHI3L1 could be a potential pharmacodynamic biomarker for VT1021 (Table 5).

**Table 4 | Peripheral TSP-1 levels and post-VT1021 fold changes by clinical responses in all evaluable patients in a phase I expansion study**

| Peripheral blood samples | Mean (95% CI) | | |
|---|---|---|---|
| | Baseline TSP-1 level | Max TSP-1 level | TSP-1 fold change |
| *PBMCs* | | | |
| CR/PR/SD ($n = 14$) | 0.29 (0.19–0.40) | 0.80 (0.41–1.20) | 3.44 (1.85–5.03) |
| PD ($n = 32$) | 0.44 (0.26–0.61) | 0.81 (0.57–1.05) | 2.67 (2.04–3.31) |
| | $p = 0.327$ | $p = 0.930$ | $p = 0.288$ |
| *Platelet* | | | |
| CR/PR/SD ($n = 14$) | 1.03 (0.57–1.48) | 1.74 (1.27–2.22) | 3.52 (1.54–5.49) |
| PD ($n = 32$) | 1.01 (0.75–1.26) | 1.50 (1.28–1.73) | 2.37 (1.68–3.07) |
| | $p = 0.873$ | $p = 0.289$ | $p = 0.200$ |
| *Plasma* | | | |
| CR/PR/SD ($n = 8$) | 2.95 (1.59–4.30) | 6.30 (4.16–8.45) | 5.41, (0.55–10.27) |
| PD ($n = 25$) | 5.50 (4.26–6.73) | 7.76 (6.00–9.52) | 1.48 (1.32–1.65) |
| | $p = 0.065$ | $p = 0.499$ | * $p = 0.037$ |

*Mean* average TSP-1 level or fold change, *CI* confidence interval, *n* the number of patients available for the analysis. Fewer plasma samples were analyzed due to sample deviation.

Baseline TSP-1 level: TSP-1 level analyzed from clinical samples collected on day 1 pre-dosing timepoint.

Max TSP-1 level: the maximum TSP-1 level from all timepoints excluding the day 1 pre-dosing timepoint.

TSP-1 Fold Change: the maximum TSP-1 level / Baseline TSP-1 level.

*CR* complete response, *PR* partial response, *SD* stable disease, *PD* progressive disease.

*Significant difference observed between patients with CR/PR/SD vs. PD; p-value calculated by unpaired *t*-test, two-tail.

**Fig. 3 | TSP-1 signal intensities in the TME from paired tumor biopsy samples before and after VT1021 treatment. a** Quantification of TSP-1 signal intensity averaged from nine patients before (screen) and after dosing with VT1021 (In-Tx). **b** Representative MIBI images from patient #24 before (screen, liver) and after dosing with VT1021 (In-Tx, liver). Screen screen biopsy sample before VT1021 treatments, In-Tx biopsy sample during or after VT1021 treatments. For MIBI images, DNA in turquoise, Keratin in purple, TSP-1 in yellow, and CD11b in red. The magnified version of the MIBI images is shown within the large red boxes. Scale bar: 21 μm in magnified images, 93 μm in unmagnified images. Error bars indicate SEM.

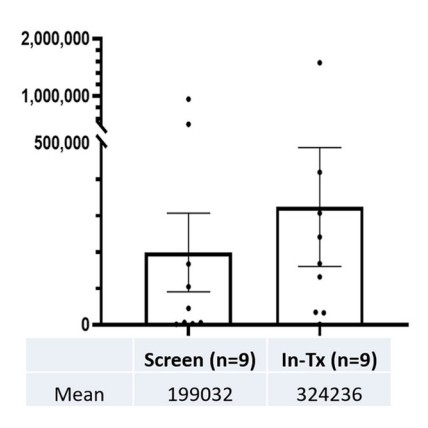

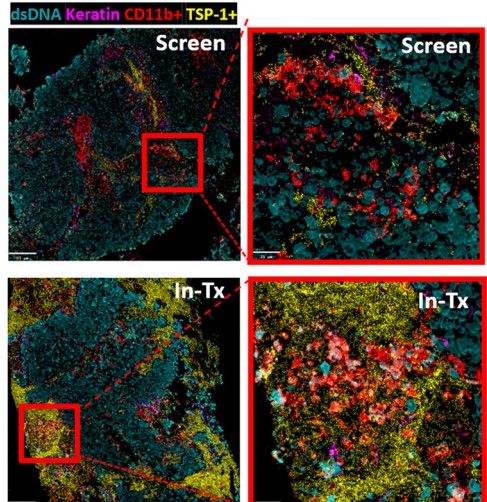

| | Screen (n=9) | In-Tx (n=9) |
|---|---|---|
| Mean | 199032 | 324236 |

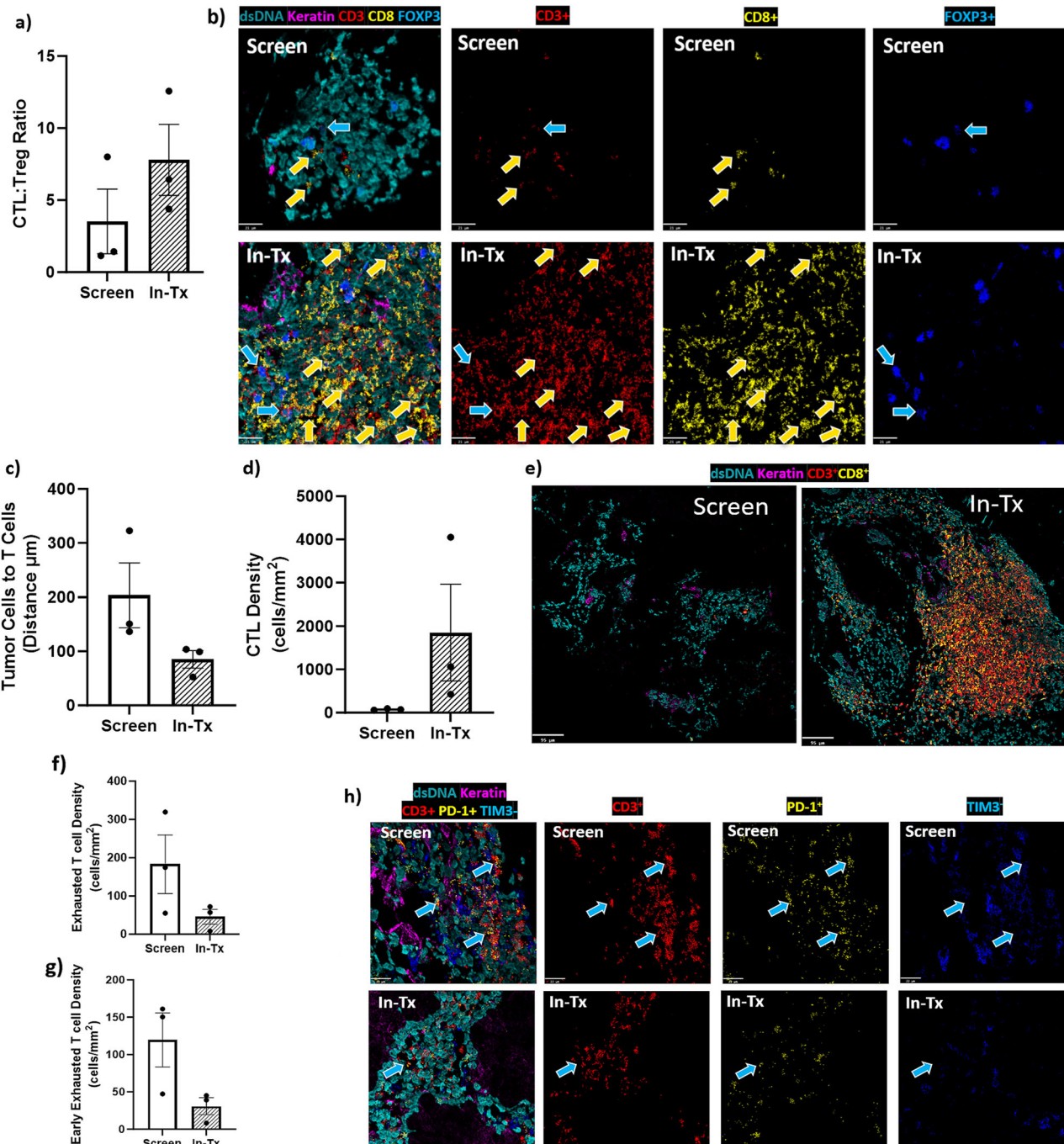

**Fig. 4 | CTL:Treg ratio, CTL to tumor distance and exhausted T cells by MIBI in paired tumor biopsy samples before and after VT1021 treatment. a–e** Data from patient #10 with ovarian cancer. **f–h** Data from patient #9 with pancreatic cancer. **a** Quantification of CTL:Treg ratio before (screen) and after dosing with VT1021 (In-Tx). **b** Representative images of tumor biopsy samples before (screen, lymph node) and after dosing with VT1021 (In-Tx, lymph node). Scale bar: 21 μm. **c** Quantification of CTL to tumor distance before (screen) and after dosing with VT1021 (In-Tx). **d** Quantification of CTL density before (screen) and after dosing with VT1021 (In-Tx). **e** Representative images of tumor biopsy samples from patient #10 before (screen, lymph node) and after dosing with VT1021 (In-Tx, lymph node). Scale bar: 95 μm. Screen: screen biopsy sample before VT1021 treatments. In-Tx: biopsy sample during or after VT1021 treatments. For MIBI images (**b** and **e**): dsDNA in turquoise, Keratin in purple, CD3⁺ in red, CD8⁺ in yellow, FOXP3⁺ in blue. Yellow Arrows show examples of CTLs which are CD3⁺ CD8⁺. Blue arrows show examples of Tregs which are CD3⁺ FOXP3⁺. **f** Quantification of total exhausted T cell densities in paired biopsy samples from lung metastases analyzed by MIBI before (screen) and after dosing with VT1021 (In-Tx). **g** Quantification of early exhausted T cell densities before (screen) and after dosing with VT1021 (In-Tx). **h** Representative MIBI images before (screen) and after dosing with VT1021 (In-Tx), dsDNA in turquoise, Keratin in purple, CD3 in red, PD-1 in yellow, TIM3 in blue. Arrows show examples of early exhausted T cells which are CD3⁺ PD-1⁺ TIM3⁻. Scale bar: 22 μm. For all bar graphs, three independent areas were used for the quantification, error bars indicate SEM.

CCL5 plays multiple roles in cancer progression and was reported to have predictive and prognostic value in several cancer types[51–53]. In our study, higher baseline CCL5 levels were observed in CR/PR/SD (33.1 ng/ml) compared to PD (23.2 ng/ml), suggesting that plasmatic CCL5 may be a predictive biomarker for VT1021 in patients with GBM and pancreatic cancer (Table 5). Notably, a 2.1–2.2-fold induction of plasmatic CCL5 was observed in all patients, regardless of clinical response, suggesting that plasma CCL5 level could be a potential

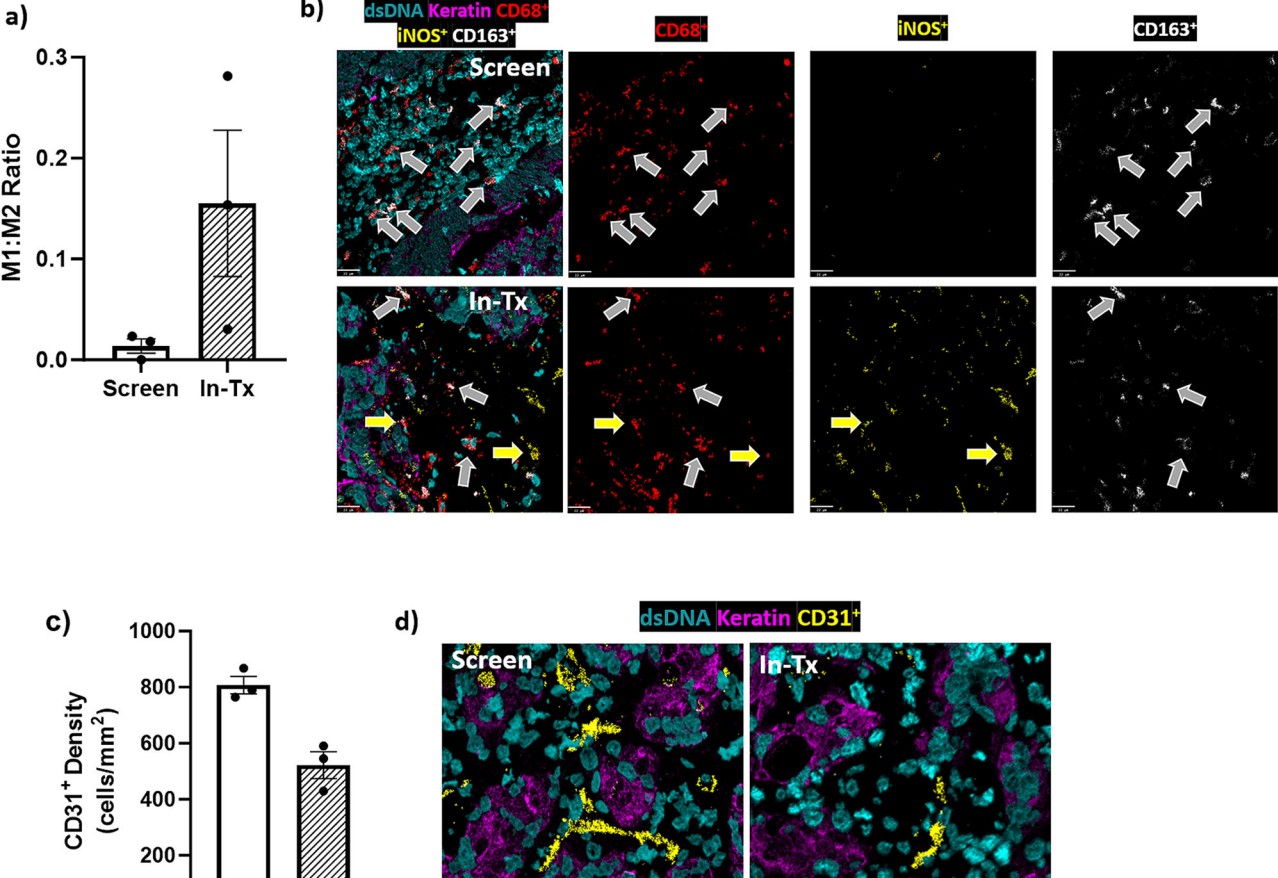

**Fig. 5 | M1:M2 ratio and microvascular density analyses by MIBI in paired tumor biopsy samples before and after VT1021 treatment in two patients enrolled in the phase I expansion study. a**, **b** Data from patient #22. **c**, **d** Data from patient #18. **a** Quantification of M1:M2 ratio from patient #22 before (screen) and after dosing with VT1021 (In-Tx). **b** Representative MIBI images for macrophages in tumor biopsy samples from patient #22 before (screen, colon) and after dosing with VT1021 (In-Tx, liver). dsDNA in turquoise, Keratin in purple, CD68 in red, iNOS in white and CD163 in yellow. Yellow arrows show examples of M1 macrophages which are CD68[+] iNOS[+]. White arrows show examples of M2 macrophages which are CD68[+] CD163[+]. Scale bar: 22 μm. **c** Quantification of CD31 from patient #18 before (screen) and after dosing with VT1021 (In-Tx). **b** Representative images of tumor biopsy samples from patient #18 before (screen, liver) and after dosing with VT1021 (In-Tx, liver). M1:M2 Ratio M1 macrophage to M2 micrphage ratio, Screen screen biopsy sample before VT1021 treatments, In-Tx biopsy sample during or after VT1021 treatments, dsDNA in turquoise, Keratin in purple, CD31 in yellow. Scale bar: 22 μm. For all bar graphs, three independent areas were used for the quantification, error bars indicate SEM.

pharmacodynamic biomarker for VT1021 (Table 5). In agreement with previous reports, patients in our study with higher CCL5 base levels and higher CCL5 induction fold may benefit from VT1021 treatment via CCL5's role to promote antitumor immunity by recruiting anti-tumor T cells and dendritic cells to the TME, thus enhancing the immunotherapy response to inhibit tumor growth[53–56].

MIF inhibits macrophage migration and is a mediator of the innate immune response that has been reported to be involved in inflammatory, autoimmune, and neoplastic disease[57,58]. In our study, a >2.0-fold reduction of MIF was observed in all patients post-VT1021 treatment, regardless of clinical response, suggesting that plasmatic MIF could be a potential pharmacodynamic biomarker for VT1021 (Table 5). Our results are consistent with previous reports that high MIF levels (tumor or serum) correlate with shorter patient survival time[58].

IL-18 Bpa is a secreted glycoprotein that functions as an IL-18 antagonist by preventing the interaction of IL-18 with IL-18 Receptor alpha[59,60]. Consequently, T cells cannot infiltrate sites of inflammation, which prevents pro-inflammatory cytokine production by CD8[+] T cells and NK cells[59,60]. In our study, a 1.5–1.9-fold reduction of IL-18 Bpa was observed in all patients post-VT1021 treatment, suggesting that plasmatic

IL-18 Bpa could be a potential pharmacodynamic biomarker for VT1021 in patients with GBM and pancreatic cancer (Table 5).

In summary, we have identified multiple cytokines as potential predictive and pharmacodynamic biomarkers for VT1021 in patients with GBM and pancreatic cancer, with PAI-1, CH13L1 and CCL5 possibly serving as both predictive and pharmacodynamic biomarkers for VT1021. However, our biomarker study is limited by the low number of clinical samples, only MMP9 showed a statistically significant difference as a predictive biomarker in this expansion study, thus additional clinical samples from future clinical trials are required to confirm these findings.

## Discussion
VT1021 is a first-in-class therapeutic agent which has been tested in a phase I clinical study in patients with solid tumors and has advanced to a phase II/III clinical study in GBM. Here, we report that VT1021 is safe and well tolerated in patients with solid tumors in a phase I expansion study. VT1021 demonstrates promising single-agent clinical activity against recurrent glioblastoma (rGBM) in this study. In addition, we provided an analysis of circulating and tissue biomarkers of VT1021 in clinical samples from evaluable patients enrolled in the expansion study. We confirmed the MOA of

**Table 5 | Plasmatic cytokine levels and post-VT1021 changes analyzed for all evaluable patients with glioblastoma and pancreatic cancer in a phase I expansion study**

| Cytokine | Baseline Mean (95% CI) | Fold change Mean (95% CI) |
|---|---|---|
| *MMP9 (ng/ml)* | | |
| CR/PR/SD (n = 7) | 93.2 (70.8, 115.6) * | −1.4 (−1.8, −1.0) |
| PD (n = 21) | 212.2 (158.7, 265.6) *, $p = 0.0205$ | −1.8 (−2.1, −1.5), $p = 0.2597$ |
| *PAI-1 (ng/ml)* | | |
| CR/PR/SD (n = 7) | 8.5 (4.2, 12.8) | −1.5 (−2.4, −0.6) |
| PD (n = 21) | 18.4 (11.5, 25.3), $p = 0.1282$ | −2.6 (−3.2, −1.9), $p = 0.106$ |
| *Chitinase-3-like-1 (ng/ml)* | | |
| CR/PR/SD (n = 7) | 47.4 (10.8, 83.9) | −2.0 (−2.6, −1.5) |
| PD (n = 21) | 64.7 (38.8, 90.5), $p = 0.5003$ | −2.4 (−3.7, −1.1), $p = 0.7687$ |
| *CCL5 (pg/ml)* | | |
| CR/PR/SD (n = 7) | 33.1 (12.6, 53.5) | 2.1 (1.4, 2.9) |
| PD (n = 21) | 23.2 (15.0, 31.4), $p = 0.2994$ | 2.2 (1.7, 2.6), $p = 0.9847$ |
| *MIF (ng/ml)* | | |
| CR/PR/SD (n = 7) | 154.8 (89.5, 220.2) | −2.1 (−2.8, −1.5) |
| PD (n = 21) | 135.4 (110.9, 159.9), $p = 0.5043$ | −2.0 (−2.3, −1.8), $p = 0.6696$ |
| *IL-18 Bpa (ng/ml)* | | |
| CR/PR/SD (n = 7) | 2.0 (1.3, 2.7) | −1.9 (−2.4, −1.4) |
| PD (n = 21) | 2.4 (1.7, 3.0), $p = 0.5436$ | −1.5 (−1.7, −1.3), $p = 0.0675$ |
| *TSP-1 (µg/ml)* | | |
| CR/PR/SD (n = 7) | 3.3 (2.1, 4.6) | 2.1 (1.7, 2.6) |
| PD (n = 21) | 5.5 (4.2, 6.7), $p = 0.0773$ | 1.5 (1.3, 1.7), *$p = 0.0103$ |

*mean* the average cytokine level, *CI* confidence interval, *n* the number of patients available for the analysis, *CR* complete response, *PR* partial response, *SD* stable disease, *PD* progressive disease. Baseline: the cytokine level analyzed from clinical samples collected on day 1 pre-dosing timepoint. Fold change: the fold change of the highest or lowest level from all other timepoints compared to the baseline. *p*: *p*-value calculated by unpaired *t*-test, two tail.
*Significant difference observed between the CR/PR/SD vs. PD.

VT1021 in a clinical setting and identified potential non-invasive biomarkers for VT1021 in future clinical studies. While the interpretation of these data is limited by the small sample size and lack of a randomized control arm, the results are encouraging and supportive of further development of VT1021 in more clinical studies.

The biological activity of VT1021 is mediated by the stimulation of TSP-1 expression[10]. TSP-1 is a large glycoprotein and interacts with more than 80 binding interactors, which mediate its complex functions[7,61]. The regulation of TSP-1 adds more complexity to its function, as it can be negatively regulated by oncogenes like Ras and Myc, as well as positively regulated by tumor suppressor genes such as p53[1,11,62]. In this study, VT1021 is designed to stimulate TSP-1 in the TME. Consistent with preclinical studies using VT1021[10], we confirmed TSP-1 induction by VT1021 in the clinical setting and our clinical biomarker data support TSP-1's role as tumor inhibiting. In this study, we found that TSP-1 levels in PBMCs and platelets in healthy donors were significantly higher than baseline TSP-1 levels in evaluable patients. Our results are consistent with previous reports showing TSP-1 levels down-regulated in various cancer types[15–19], supporting a role of TSP-1 in tumor inhibition. However, there are some contradictory reports regarding the correlation of TSP-1 expression and tumor progression[2,6,7,61,63]. Discrepancies between TSP-1 and its role in tumor progression may be attributed to previous studies using different model systems, cancer indications, tumor stages, varying

types of samples, or measuring cellular TSP-1. For instance, cellular TSP-1 expression level has been analyzed comprehensively in glioma with three data sets, TCGA, CGGA and GSE[64]. This study showed that cellular TSP-1 (RNA or protein) positively correlates with malignancy. As TSP-1 is a secreted protein, its abundance in the TME depends on both tumor cells and stromal cell expression, and it is unlikely to be reflected by the measurement of TSP-1 intracellular levels alone. Our biomarker data suggests that secreted TSP-1 induction is positively correlated with clinical responses, supporting a positive role of secreted TSP-1 in tumor inhibition.

Our MIBI data supports the role of TSP-1 in reprogramming the TME to inhibit tumor growth. In this phase I expansion study for VT1021, only nine biopsy pairs were available for MIBI analysis. We analyzed the TME in those patients and observed TME modifications post-VT1021 in a subset of patients, such as increased CTL:Treg ratio, decreased CTL to tumor distance, decreased T cell exhaustion, increased M1:M2 ratio and decreased MVD. These changes in the TME support VT1021 in reprogramming the TME and inhibiting tumor growth. Although only a subset of patients that we analyzed showed these TME changes, we attribute this to a few possibilities: (1) Selected ROIs for MIBI analysis only reflect <10% of the biopsy sample, not the entire biopsy sample; (2) Biopsy sample variations due to sample acquisition time, biopsy types, and sample processing variations; (3) In all cases the screen biopsy was not taken from the same lesion that was used to obtain the in-treatment biopsy. Therefore, it is difficult to ascertain if the screen lesion would have responded in the same manner following VT1021 treatment.

Non-invasive predictive biomarkers are valuable for patient stratification in clinical drug development. In this study, we found that plasmatic MMP9 levels may be a potential predictive biomarker for VT1021, which is consistent with data from a phase 3 study[42]. Moreover, MMP9 has been shown to be inhibited by TSP-1 and thus may be directly linked to the biological activity of VT1021. Additional cytokines identified in this study could be potential predictive or pharmacodynamic biomarkers for VT1021, more clinical samples from future clinical trials are required to confirm these findings.

Taken together, we report the safety and tolerability, clinical response, and biomarker profile of VT1021 in the phase I expansion study. We report the clinical confirmation of TSP-1 induction by VT1021 in peripheral blood samples and the TME. We observed modifications of the TME that correlated with VT1021 treatment, from one that is immunosuppressive and tumor-promoting to one that is immune active and tumor-inhibiting. Our data supports the biological MOA of VT1021, a first-in-class therapeutic agent, in stimulating TSP-1 and reprogramming the TME to inhibit tumor growth.

## Data availability
All source data for the figures can be found in Supplementary Data. The study protocol is available within the Supplementary Information of reference 23. Individual participant data that underlies the results reported in this paper after deidentification will be shared upon request. Data requests by researchers with reasonable proposals will be accepted immediately following publication and ending 5 years following publication.

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

## Acknowledgements
The authors thank the patients and their families for their effort and dedication in participating in this study during the COVID-19 pandemic. Thanks to Harold Pestana, Judy Chiao, Lou Vaickus, Manmeet Ahluwalia, Marsha Crochiere, William Craig, Robert Guttendorf, and Aurelia Mickunas for their advice and support in the clinical trial and sample coordination.

## Author contributions
Patient enrollment and study monitoring: D.S., D.P., D.M., J.B., M.R.P., D.J., P.Y.W., A.B, J.E.S., S.P., and J.L. Clinical data acquisition and analysis: J.J.C, J.W. Biomarker data acquisition and analysis: J.J.C., M.Y.V., J.W., W.L., S.F., V.Z., R.S.W. Clinical sample coordination: W.L., M.Y.V., S.W. Data interpretation: J.J.C., J.W., R.S.W., J.M., M.C. study supervision: J.J.C., J.W.

## Competing interests
Vigeo Therapeutics designed VT1021 and sponsored the trial in this article. R.S.W. is a co-founder of, and consultant for, Vigeo Therapeutics, which has licensed technology from Boston Children's Hospital. V.Z. is a consultant for Vigeo Therapeutics. J.J.C., M.Y.V., W.L., S.F., S.W., J.M., M.C., and J.W. are employees of Vigeo Therapeutics. D.S., D.P., D.M., J.B., M.R.P., D.J., P.Y.W., A.B, J.E.S., S.P., and J.L. declare no competing interests.

## Additional information

**Peer review information** : *Communications Medicine* thanks Jan Rekowski and the other, anonymous, reviewer(s) for their contribution to the peer review of this work. A peer review file is available

**Article**

**Jian Jenny Chen** [1] ✉**, Melanie Y. Vincent**[1]**, Dale Shepard**[2]**, David Peereboom**[2]**, Devalingam Mahalingam** [3]**, James Battiste**[4]**, Manish R. Patel**[5]**, Dejan Juric**[6]**, Patrick Y. Wen**[7]**, Andrea Bullock** [8]**, Jennifer Eva Selfridge**[9]**, Shubham Pant**[10]**, Joyce Liu**[7]**, Wendy Li**[1]**, Susanne Fyfe**[1]**, Suming Wang**[1]**, Victor Zota**[1]**, James Mahoney**[1]**, Randolph S. Watnick**[11]**, Michael Cieslewicz**[1] **& Jing Watnick** [1] ✉

[1]Vigeo Therapeutics, Cambridge, MA, USA. [2]Cleveland Clinic, Cleveland, OH, USA. [3]Northwestern Memorial Hospital, Chicago, IL, USA. [4]Stephenson Cancer Center, Oklahoma City, OK, USA. [5]Florida Cancer Specialists/Sarah Cannon Research Institute, Sarasota, FL, USA. [6]Massachusetts General Hospital, Boston, MA, USA. [7]Dana Farber Cancer Institute, Boston, MA, USA. [8]Beth Israel Deaconess Medical Center, Boston, MA, USA. [9]University Hospitals Cleveland Medical Center, Cleveland, OH, USA. [10]The University of Texas MD Anderson Cancer Center, Houston, TX, USA. [11]Boston Children's Hospital, Boston, MA, USA. ✉e-mail: jenny.chen@vigeotx.com; jing.watnick@vigeotx.com

