## [Peer Review File · Communications Medicine]

Reviewers' comments:

Reviewer #1 (Remarks to the Author):

Chen et al, have investigated comprehensive analysis of VT1021, and found that it induces TSP-1 levels in peripheral blood and tumor biopsy samples. Furthermore, authors had observed that treatment with VT1021 led in the remodeling of the Tumor Microenvironment by alternating immunosuppressive and tumor-promoting to immune active and tumor-inhibiting phenotype. This is due to increase in ratio of CTL:Treg ratio, M1:M2 macrophage ratio, Tumor-infiltrating lymphocytes, and reducing T cell exhaustion and microvascular density. Authors have validated MMP9, PAI-1, CHI3L1, CCL5 MIF, IL-18 Bpa as non-invasive biomarkers for VT1021 activity. Authors have concluded CHI3L1, CCL5 MIF and IL-18 Bpa could be potential pharmacodynamic biomarkers for VT1021. Overall, the manuscript is well written, and authors have provided copious amount of supplementary and protocol information related with manuscript. There are few clarifications that needs to be addressed for better understanding of study and readability of the manuscript as described below:

1. Please insert reference or citation for method used to process Blood samples for ELISA and quantitative RT-PCR (qRT-PCR) assays.
2. Please clarify what type of samples are used for baseline to calculate induction fold of TSP1 in Figure1. Are these change in fold significant? No asterisks are shown for p-value in the graphs Figure 1, A-D?
3. Figure2, illustration of lines for significant p-value are confusing. Panel C, healthy donors vs In-Tx subjects are significant?
4. Please clarify baseline for Figure 3 and define in the method section or legends clearly.
5. Please specify 'Screen' means/reference in legends or method section.
6. Are 'subject #' means number of patients or given number to particular patient's sample. Please clarify.
7. Figure 9, MMP9 is shown as significant, but speculated other potential biomarker CHI3L1, CCL5 MIF and IL-18 Bpa for VT1021 treatment. It seems that Figure 9, does not agree with authors conclusions. Authors should clarify.
8. Authors should show level of TSP1 along with MMP9 in Figure 9. TSP1 has been reported to spontaneous tumor growth and inhibits activation of MMP-9 and mobilization of vascular endothelial growth factor (PMID: 11606713).

Reviewer #2 (Remarks to the Author):

In this article, Chen JJ and colleagues evaluated the impact of a peptide, VT1021, known to increase TSP1 expression in the tumor microenvironment, in a clinical study on a plethora of patients. Patients were chosen based on aggressive tumors, such as recurrent glioblastoma, metastatic ovarian cancer.

This is an interesting study, well designed, many interesting information can be extracted from it. I have some questions before considering the article ready for being accepted.

1/ Authors discuss the controversial role of TSP1 in anti- or pro-tumoral properties, but it seems that VT1021 increases TSP1 expression in tumor cells, which was shown to worsen survival in several preclinical studies. Do the authors think that the TSP1 secreted by immune cells (e.g) is different

than the one secreted by tumor cells ?

2/ It was reported that TSP1 can be cleaved, did the authors observe this cleaved form ? Can the ELISA also detect this cleaved form ? Can the authors perform some WB on PDMC of patients to see the TSP1 profile?

3/ GBM patient characteristics were not included in the “Table 1. Baseline patient demographics and characteristics, by treatment group », please add it.

4/ Can the author add survival curves of the treated patient to compare to normal survival curves of untreated patients (Kaplan-Meier).

Reviewer #3 (Remarks to the Author):

Summary. The study reported TSP-1 induction and TME biomarkers for the association with VT120. Unclear trial design and discrepancy of sample size make difficult to assess the biomarker analysis of this manuscript.

- Unclear trial design and discrepancy of sample size. Evaluation of biomarker analysis in this manuscript used the expansion cohort with n=46 (among them, n=38 for response association). In contrast, the attached manuscript primary for clinical trial result indicated a dose escalation + expansion cohorts with n=38 (note not clear which is the dose escalation cohort and which one is the expansion cohort) and only 28 patients for response evaluation. The conflicting description makes difficult to assess the biomarker analysis of this manuscript.
- What was the trend of the TSP-1 biomarker over time (15 time points)? What is the rationale to use maximum or minimum analyte levels for fold change calculation and in-treatment comparison (Fig 1-2)? How to determine when to use the max or min metric?
- Any difference of TSP-1 induction fold post-VT1021 among groups (GBM, pancreatic, and others) in Fig 1?
- What were statistical methods used to compare pre- and post-treatment in Fig 2? Did the p value be adjusted for multiple comparison? Same question for Fig 3.
- The clinical trial had various secondary endpoints, such as ORR, PFS, in addition to DCR. Biomarker analysis should utilize these clinical endpoints.
- TME analysis should be indicated as ‘descriptive analysis’ given the limited sample size (n=9).

Reviewer #4 (Remarks to the Author):

This manuscript reports the biomarker analyses from a phase I/II trial in solid tumours. The authors claim to have confirmed in a clinical setting the mechanism of action of VT1021, a TSP-1 expression stimulating amino acid peptide. They report to have observed a change in tumour microenvironment after treatment with VT1021 from immunosuppressive and tumour-promoting to immune-active and tumour-inhibiting. They also suggest several putative predictive/prognostic biomarkers for VT1021 but caution more research is needed.

The manuscript has a clear structure and is well-written. However, interpretation of the results suffers from the fact that clinical characteristics for this patient population have not yet been fully reported. The authors provide the cancer type and best response to treatment when singling out

patients, which is very much appreciated, but there is no Table 1 with clinical characteristics of the patient population and secondary outcomes beyond best response are not presented either. Linking clinical with biomarker outcomes is especially key for the interpretation of the results if only a subgroup of patients provided biopsies; it is important to know which patients were included in these analyses. You would, for example, want to know whether the mechanism of action seen in the pharmacodynamic analyses also translated into longer duration of response or clinical benefit, or whether a larger increase/decrease in key parameters resulted in a larger reduction in tumour size (e.g., in a waterfall plot). Although there are no clear-cut rules on this matter, this reviewer's opinion is that reporting clinical and biomarker analyses together would have provided a more complete picture and, in fact, resulted in a much stronger manuscript. If presented separately, I would argue the clinical analysis should precede the biomarker analysis to be properly referenced in the latter.

There are some specific comments, especially regarding statistical analyses, that might help to improve the manuscript:

1. My biggest concern relates to the "quantification of selected ROIs" and the associated statistical tests. The t-test like almost all standard statistical tests assumes independence of observational units. It appears unlikely to me that regions of interest from the same sample can be considered independent. It is also not advised to select one patient based on a large difference in a certain parameter to then test for this very same difference statistically within this patient. Instead of disregarding all the other patients' data, a reasonable approach would be to test this hypothesis using all available data accounting for the correlation between regions of interest from the same sample/patient. The effect measure of interest would then be the average x-fold increase in that parameter, and a confidence interval could be provided too. In addition, a bar plot is not needed if there are only three data points.
2. These analyses, however, are further undermined by the limitations stated in lines 388-390, especially that the screening biopsy and the in-treatment biopsy were not from the same lesion. A direct comparison may thus be biased or even falsified by differences due to, e.g., biopsy type and location and/or selection of ROIs.
3. There is limited data available for paired analyses, i.e., for 9 out of 46 patients in total. Several of the paired analyses are then only available for a further subset of these 9 patients. A patient flow chart would be very helpful to see which patients contributed to what analyses.
4. In lines 82/83, the authors state that "[a]ll patients received VT1021 (11.8mg/kg, twice a week intravenously)". I assume this relates to the expansion phase (see line 80), which as far as I understand comprised 46 patients (cf. Figure 1B). However, reference #23 reports that "[i]n the escalation phase, 46 subjects received between 0.5–15.6 mg/kg of VT1021 by IV infusion twice weekly". Thus, it is not clear to me whether the 46 patients reported here are from the escalation phase, the expansion phase, or both, and at what dose they were treated.
5. In line 97, the authors state that "a pre-study or archival tumor specimen was collected". Can the authors confirm this sample was taken after the patient signed the consent form? The wording is somewhat ambiguous as archival could also relate to a sample taken at the time of initial diagnosis before any previous lines of treatment.
6. The authors show boxplots with median values and outliers throughout the manuscript. However, there is no definition of what constitutes an outlier (which can be different from one statistical software package to another).
7. Regarding boxplots, the authors might want to consider a combination of violin and dot plots instead to allow the reader to assess entire distribution of a parameter. The tables indicating count, median and outliers could then be omitted with "count" being incorporated in the x-axis text (i.e., "CR/PR/SD (n=7)") and exceptionally interesting median values being stated in the main text.

8. ANOVA has been used for between-group comparisons. For some parameters, there are clear outliers which can bias the results of an ANOVA. Non-parametric approaches to statistical inference like the Mann-Whitney U test (for two groups) or the Kruskal-Wallis test (for more than two groups) would provide a more robust choice. This would also be in line with showing the median (a non-parametric measure) instead of the arithmetic mean. An alternative option would be to completely abstain from statistical significance testing for these exploratory analyses and give more emphasis to estimation and precision (that is, effect measures and their confidence intervals).
9. In Figure 3, the authors present a total of 9 analyses highlighting the significant difference in secreted TSP-1 induction between CR/PR/SD and PD patients. Given that they do not seem to have corrected for multiple testing, this result may be due to chance and thus the conclusions drawn in lines 192-195 appear too strong.
10. In line 303, what do the authors mean with "more robust" fold changes? In general, it is not entirely clear according to what criteria these four biomarkers were selected from the 105 cytokines panel.
11. In line 398, the authors state that "[a]dditional cytokines (...) showed a trend (...) to be biomarkers". Trends in the statistical sense can either be with regard to time or across the levels of an ordinal variable. Neither is the case here, so the term trend should be avoided.
12. The quality of the figures is suboptimal in general. Vertical axes should have the same scale if they are measuring the same (e.g., in Figure 1). Instead of showing asterisks to indicate p-values, the actual p-values should be shown. Figure 4C/4F don't seem to be the best way to represent these data (if a graphical representation is required at all).

Dear all,

In response to reviewers' comments received on 6/29/2023 on our manuscript (COMMSMED-23-0132-A), we have added clinical data with the additional tables (Table 1 &2) and figure (Figure 1) to this manuscript. Please note that we have listed additional PIs (principal investigators) as authors and revised the title to reflect this change. The subject ID numbers in the biomarker studies have been revised to be consistent with the newly added clinical data.

For your convenience, we have made a list of Tables and Figures of the revised version in comparison with the originally submitted version.

Manuscript: COMMSMED-23-0132-A

First version (Feb 2023)	Revised version (August 2023)
Title: Biomarker analysis of VT1021, a first-in-class therapeutic agent that stimulates Thrombospondin-1 and reprograms the tumor microenvironment	Title: Clinical biomarker analyses of a phase 1 expansion study assessing VT1021, a first-in-class immunotherapy agent in patients with solid tumors
Authors: Jian Jenny Chen, *, Melanie Y. Vincent, Wendy Li, Susanne Fyfe, Suming Wang, Victor Zota, James Mahoney, Randolph Watnick, Michael Cieslewicz and Jing Watnick, *	Authors: Jian Jenny Chen, *, Melanie Y. Vincent, Dale Shepard, David Peereboom, Deva Mahalingam, James Battiste, Manish R. Patel, Dejan Juric, Patrick Y. Wen, Andrea Bullock, Jennifer Eva Selfridge, Shubham Pant, Joyce Liu, Wendy Li, Susanne Fyfe, Suming Wang, Victor Zota, James Mahoney, Randolph Watnick, Michael Cieslewicz and Jing Watnick, *
NA	Table 1. Patient demographics and baseline characteristics
NA	Figure 1. Swimmer plot showing best clinical responses
NA	Table 2. Related Treatment-Emergent Adverse Events
Figure 1. TSP-1 induction fold	Figure 2. TSP-1 induction fold
Figure 2. TSP-1 in healthy donors and subjects	Table 3. TSP-1 in healthy donors and subjects
Figure 3. TSP-1 by clinical responses	Table 4. TSP-1 by clinical responses
Figure 4. TSP-1 in TME	Figure 3. TSP-1 in TME
Figure 5. CTL: Treg ratio and CTL to tumor distance	Figure 4. CTL: Treg ratio and CTL to tumor distance
Figure 6. Exhausted T cells	Figure 5. Exhausted T cells
Figure 7. M1:M2 ratio	Figure 6. M1:M2 ratio
Figure 8. Microvascular density	Figure 7. Microvascular density
Figure 9. Plasmatic cytokine levels	Table 5. Plasmatic cytokine levels

Referee expertise:

Referee #1: TSP-1, TME, angiogenesis

Referee #2: TSP-1, TME, angiogenesis

Referee #3: Statistics, biomarkers, trials

Referee #4: Statistics, biomarkers, trials

Reviewers' comments (6/29/2023):

Reviewer #1 (Remarks to the Author):

Chen et al, have investigated comprehensive analysis of VT1021, and found that it induces TSP-1 levels in peripheral blood and tumor biopsy samples. Furthermore, authors had observed that treatment with VT1021 led in the remodeling of the Tumor Microenvironment by alternating immunosuppressive and tumor-promoting to immune active and tumor-inhibiting phenotype. This is due to increase in ratio of CTL:Treg ratio, M1:M2 macrophage ratio, Tumor-infiltrating lymphocytes, and reducing T cell exhaustion and microvascular density. Authors have validated MMP9, PAI-1, CHI3L1, CCL5 MIF, IL-18 Bpa as non-invasive biomarkers for VT1021 activity. Authors have concluded CHI3L1, CCL5 MIF and IL-18 Bpa could be potential pharmacodynamic biomarkers for VT1021. Overall, the manuscript is well written, and authors have provided copious amount of supplementary and protocol information related with manuscript. There are few clarifications that needs to be addressed for better understanding of study and readability of the manuscript as described below:

1. Please insert reference or citation for method used to process Blood samples for ELISA and quantitative RT-PCR (qRT-PCR) assays.

>> We have inserted the reference and citation for the method used to process Blood samples for ELISA and quantitative RT-PCR (qRT-PCR) assays, please see page 4.

2. Please clarify what type of samples are used for baseline to calculate induction fold of TSP1 in Figure1. Are these change in fold significant? No asterisks are shown for p-value in the graphs Figure 1, A-D?

>> We have labeled sample types in the revised Figure (now Figure 2) with a table under each graph, and we clarified in the text (page 7). The fold changes are not significant between different cancer types, indicating that TSP-1 induction by VT1021 has been observed for all cancer types tested in this study. We have clarified this in the revised manuscript (page 7).

3. Figure2, illustration of lines for significant p-value are confusing. Panel C, healthy donors vs In-Tx subjects are significant?

>> We thank the reviewer for bringing this to our attention. For Panel C (plasma), healthy donors vs In-Tx subjects are significant. The p-values are 0.001, 0.0003, and 0.0206 for GBM, pancreatic cancer and other cancer types, respectively. We have included the above p-values in the revised Table (Table 3 to replace Figure 2).

In the revised version, to reflect the difference between all the groups, we replaced Figure 2 with Table 3 and provided the mean value and 95% CI (confidence interval) for each group. We have included all

the p-values that show significant difference ($p < 0.05$) and two p-values approaching significant difference ($0.05 < p < 0.06$).

4. Please clarify baseline for Figure 3 and define in the method section or legends clearly.

>> We have clarified baseline for Figure 3 (replaced by Table 4) in the revised manuscript and defined in the method section (page 4).

5. Please specify 'Screen' means/reference in legends or method section.

>> We have specified "Screen" in the method section under "Clinical trial and sample acquisition" (Page 5).

6. Are 'subject #' means number of patients or given number to particular patient's sample. Please clarify.

>> The 'subject #' is a given number to a particular patient. We have clarified in the revised version (page 4). A subject ID number is an artificial number assigned to each patient enrolled in this study to protect the patient's privacy. The subject ID number used in this manuscript is not the patient ID number used in clinical sites.

7. Figure 9, MMP9 is shown as significant, but speculated other potential biomarker CHI3L1, CCL5 MIF and IL-18 Bpa for VT1021 treatment. It seems that Figure 9, does not agree with authors conclusions. Authors should clarify.

>> We agree with the reviewer; we have stated that "However, our biomarker study is limited by the low number of clinical samples, only MMP9 showed a statistically significant difference as a predictive biomarker in this expansion study, thus additional clinical samples from future clinical trials are required to confirm these findings." (Page 12).

8. Authors should show level of TSP1 along with MMP9 in Figure 9. TSP1 has been reported to spontaneous tumor growth and inhibits activation of MMP-9 and mobilization of vascular endothelial growth factor (PMID: 11606713).

>> We thank the reviewer for the suggestion. In the revised version, we replaced Figure 9 with a table (Table 5) and showed levels of all cytokines including TSP-1.

As pointed out by the reviewer, TSP-1 has been reported to suppress spontaneous tumor growth and inhibit activation of MMP9. With this clinical biomarker study, we were not able to show direct inhibition of MMP9 activation by TSP-1, we can only observe the correlation between TSP-1 and MMP9 after VT1021 treatments.

As shown in Table 5, we have observed a 1.5-2.1-fold up-regulation of TSP-1 in subjects with GBM and pancreatic cancer after VT1021 treatment, and a 1.4-1.8-fold reduction of MMP9 in these subjects. We have observed upregulation of TSP-1 upon VT1021 treatments and down-regulation of MMP9 after VT1021 treatment in the same patient population, supporting the role of TSP-1 in suppressing MMP9.

Reviewer #2 (Remarks to the Author):

In this article, Chen JJ and colleagues evaluated the impact of a peptide, VT1021, known to increase TSP1 expression in the tumor microenvironment, in a clinical study on a plethora of patients. Patients were chosen based on aggressive tumors, such as recurrent glioblastoma, metastatic ovarian cancer. This is an interesting study, well designed, many interesting information can be extracted from it. I have some questions before considering the article ready for being accepted.

1/ Authors discuss the controversial role of TSP1 in anti- or pro-tumoral properties, but it seems that VT1021 increases TSP1 expression in tumor cells, which was shown to worsen survival in several preclinical studies. Do the authors think that the TSP1 secreted by immune cells (e.g) is different than the one secreted by tumor cells?

>> Yes, this is what we think. It has been reported that TSP-1 produced by myeloid-derived suppressor cells (immune cells) contributed to its tumor-inhibitory function, we have included the reference in the manuscript (ref #22). In this clinical study, we observed the upregulation of TSP-1 in MDSCs after VT1021 treatment (page 8-9).

2/ It was reported that TSP1 can be cleaved, did the authors observe this cleaved form? Can the ELISA also detect this cleaved form? Can the authors perform some WB on PDMC of patients to see the TSP1 profile?

>> We performed the WB (SDS-page) on PBMCs and we observed additional forms other than the full-length TSP-1 (see below for an example). The ELISA kit that we used can detect all the forms of TSP-1. The 150-180KD bands should be the full-length TSP-1, and the 120KD form is likely the “cleaved form” of TSP-1. Anastasi and co-workers (Science signaling, 2020) reported that Bone morphogenetic protein 1 (BMP-1) cleaves TSP-1 and yields a 120KD form. It is not feasible to test the hypothesis with clinical samples, more *in vitro* experiments will be required to investigate how this 120KD form is produced.

3/ Baseline patient demographics and characteristics, by treatment group, please add it.

>> We thank the reviewers for the suggestion, we have added Table 1 for patient demographics and baseline characteristics. This is a single arm phase 1 expansion study with one treatment group which was VT1021. We are sorry for the confusion; and we have clarified this in the revised manuscript (Title, Plan language summary and Abstract (page 1)).

4/ Can the author add survival curves of the treated patient to compare to normal survival curves of untreated patients (Kaplan-Meyer).

>> This is a single arm phase 1 expansion study; we are not able to compare survival curves as suggested.

Reviewer #3 (Remarks to the Author):

Summary. The study reported TSP-1 induction and TME biomarkers for the association with VT120. Unclear trial design and discrepancy of sample size make difficult to assess the biomarker analysis of this manuscript.

- Unclear trial design and discrepancy of sample size. Evaluation of biomarker analysis in this manuscript used the expansion cohort with n=46 (among them, n=38 for response association). In contrast, the attached manuscript primary for clinical trial result indicated a dose escalation + expansion cohorts with n=38 (note not clear which is the dose escalation cohort and which one is the expansion cohort) and only 28 patients for response evaluation. The conflicting description makes difficult to assess the biomarker analysis of this manuscript.

>> This manuscript only covers the phase 1 expansion study. We are sorry for the confusion; and we have clarified this in the revised manuscript (Title, Plan language summary and Abstract (page 1)).

In the revised version, we have added a swimmer plot (Figure 1) to show clinical outcomes of all subjects enrolled in this phase 1 expansion study. A total of 47 evaluable subjects have been enrolled in this expansion study. One subject had day 1 samples damaged, therefore the baseline biomarker data is only available for 46 subjects. We have included this information in the revised manuscript (page 7).

To clarify, the phase I escalation study has been submitted separately to the same Journal (Communications Medicine) and it has been accepted in September 2023.

- What was the trend of the TSP-1 biomarker over time (15 time points)? What is the rationale to use maximum or minimum analyte levels for fold change calculation and in-treatment comparison (Fig 1-2)? How to determine when to use the max or min metric?

>> We thank the reviewer for asking this question. When we designed the study, our goal was to identify a pattern of changes in TSP-1 (or other cytokines) in response to VT1021 treatments across 15 timepoints. However, at the end of the study, we decided to use maximum or minimum analyte levels for fold change calculation and in-treatment comparison for two reasons:

- 1) This phase 1 clinical study was initiated just days prior to the declaration of the pandemic. In the Methods section we note that sample deviations (30%) occurred due to the COVID-19 pandemic (2019-2021), some samples were not collected at clinical sites as scheduled or some samples were damaged during the shipping process (Page 4-5).
 - a. Some clinical sites were unable to keep cancer patients in the hospital after VT1021 infusion to limit potential exposure to COVID virus; in those cases, we were only able to collect pre-dose (pre-) and VT1021 post-infusion (0 hour) samples. Therefore, we were able

to have pre-dosing samples for baseline biomarker analysis but missed samples at later timepoints.

- b. Ongoing labor shortages at both FedEx or UPS during the pandemic led to a loss of samples during the shipping process. Sample delivery for processing was sometimes delayed 2-4 days and the sample deviation was random regarding timepoints for each patient. Due to the random nature of sample loss impacted by Covid19 pandemic we were unable to analyze all samples across all time points.

2) With the available samples, the trend of TSP-1 changes is not clear across 15 timepoints as shown in the graph below (plotted with TSP-1 mean values, error bars indicate SEM, n refers to the number of data points).

Although we can't see an obvious trend in TSP-1 changes across 15 timepoints, we observed significant TSP-1 changes after 50 days of treatment with VT1021. When we grouped all TSP-1 data points in Day 1 and Day 4, then compared to all TSP-1 data points in Day 53, we have seen significant up-regulation of TSP-1 as shown in the graph below (plotted with TSP-1 mean values, error bars indicate SEM, n refers to the number of data points).

Regarding when to use the Max or Min. When we plot all available datapoints across time regardless of cancer type or clinical response, the direction of cytokine changes (reduction or induction) after VT1021 can be observed. See above graphs for two examples: induction of TSP-1 and reduction of MMP9 overtime.

Since sample collections for patients were missing randomly regarding timepoints and we can't define which timepoint to use due to lack of a clear trend, we use Max or Min to show the highest capability of VT1021 to induce or reduce a cytokine.

- Any difference of TSP-1 induction fold post-VT1021 among groups (GBM, pancreatic, and others) in Fig 1?

>> No significant difference observed among the groups; we have included this information in the revised version (page 7).

- What were statistical methods used to compare pre- and post-treatment in Fig 2? Did the p value be adjusted for multiple comparison? Same question for Fig 3.

>> In the revised version, as suggested by reviewers, we reconsider the statistical methods and have replaced Figure 2 with Table 3 and provided the mean values and confidence intervals for all groups. Similarly, we have replaced Figure 3 with Table 4. We have added a "Statistical analysis" section under the Methods to explain how we calculate the p-value (page 6). Additional information was added to each table/figure under the legend.

- The clinical trial had various secondary endpoints, such as ORR, PFS, in addition to DCR. Biomarker analysis should utilize these clinical endpoints.

>> The original design of the Phase 1 expansion study was to enroll up to 15 subjects in each of solid tumor indications: ovarian cancer (Cohort A), pancreatic cancer (Cohort B), triple negative breast cancer (cohort C), glioblastoma (Cohort D) and basket cohort of patients with high levels in CD36 (Cohort E) for a total of 85 subjects. Once the pandemic was declared, there were challenges to enrolling patients due to local hospital restrictions. Cohorts A, C and E were severely under-enrolled. The lack of enrollment in each indication made it difficult to perform correlative analysis between PFS and ORR to biomarkers. We do believe that the correlative analysis between clinical responses (CR, PR, SD and PD) and biomarker profile is meaningful in demonstrating the MoA of VT1021.

- TME analysis should be indicated as 'descriptive analysis' given the limited sample size (n=9).

>> We agree with the reviewer and specified the TME analysis as 'descriptive analysis' in the revised version under the Abstract (page 2) and Results (page 8).

Reviewer #4 (Remarks to the Author):

This manuscript reports the biomarker analyses from a phase I/II trial in solid tumours. The authors claim to have confirmed in a clinical setting the mechanism of action of VT1021, a TSP-1 expression stimulating amino acid peptide. They report to have observed a change in tumour microenvironment after treatment with VT1021 from immunosuppressive and tumour-promoting to immune-active and tumour-inhibiting. They also suggest several putative predictive/prognostic biomarkers for VT1021 but caution more research is needed.

The manuscript has a clear structure and is well-written. However, interpretation of the results suffers

from the fact that clinical characteristics for this patient population have not yet been fully reported. The authors provide the cancer type and best response to treatment when singling out patients, which is very much appreciated, but there is no Table 1 with clinical characteristics of the patient population and secondary outcomes beyond best response are not presented either. Linking clinical with biomarker outcomes is especially key for the interpretation of the results if only a subgroup of patients provided biopsies; it is important to know which patients were included in these analyses. You would, for example, want to know whether the mechanism of action seen in the pharmacodynamic analyses also translated into longer duration of response or clinical benefit, or whether a larger increase/decrease in key parameters resulted in a larger reduction in tumour size (e.g., in a waterfall plot). Although there are no clear-cut rules on this matter, this reviewer's opinion is that reporting clinical and biomarker analyses together would have provided a more complete picture and, in fact, resulted in a much stronger manuscript. If presented separately, I would argue the clinical analysis should precede the biomarker analysis to be properly referenced in the latter.

>> We thank the reviewer for the suggestions. We have added Table 1 for patient demographics and baseline characteristics. In addition, we added a swimmer plot (Figure 1) to show clinical outcomes of all 47 evaluable subjects enrolled in this phase 1 expansion study.

There are some specific comments, especially regarding statistical analyses, that might help to improve the manuscript:

1. My biggest concern relates to the "quantification of selected ROIs" and the associated statistical tests. The t-test like almost all standard statistical tests assumes independence of observational units. It appears unlikely to me that regions of interest from the same sample can be considered independent. It is also not advised to select one patient based on a large difference in a certain parameter to then test for this very same difference statistically within this patient. Instead of disregarding all the other patients' data, a reasonable approach would be to test this hypothesis using all available data accounting for the correlation between regions of interest from the same sample/patient. The effect measure of interest would then be the average x-fold increase in that parameter, and a confidence interval could be provided too. In addition, a bar plot is not needed if there are only three data points.

>> We thank the reviewer for the suggestions. Due to the low number (n=9) of available paired biopsy samples, the statistical analysis may not reflect the biological significance of our observation. We have revised the manuscript and specified the TME analysis as 'descriptive analysis' in the revised version under the Abstract (page 2) and Results (page 8).

2. These analyses, however, are further undermined by the limitations stated in lines 388-390, especially that the screening biopsy and the in-treatment biopsy were not from the same lesion. A direct comparison may thus be biased or even falsified by differences due to, e.g., biopsy type and location and/or selection of ROIs.

>> Due to known reasons, the screening biopsy and the in-treatment biopsy were not able to be taken from the same lesion. This limitation is universal for all the paired biopsy research from any clinical studies. Due to low numbers of paired biopsy samples and above-mentioned limitations, we have revised the manuscript indicating that TME analysis as 'descriptive analysis' (page 2 & 8).

3. There is limited data available for paired analyses, i.e., for 9 out of 46 patients in total. Several of the paired analyses are then only available for a further subset of these 9 patients. A patient flow chart would be very helpful to see which patients contributed to what analyses.

>> In the revised version, we have added a swimmer plot (Figure 1) to show clinical responses of all evaluable subjects enrolled in this phase 1 expansion study. Subject ID numbers are shown for all subjects (patients) in Figure 1 and referenced later in the biomarker studies. The biomarker data from each patient can be correlated with their clinical outcomes shown in Figure 1.

4. In lines 82/83, the authors state that "[a]ll patients received VT1021 (11.8mg/kg, twice a week intravenously)". I assume this relates to the expansion phase (see line 80), which as far as I understand comprised 46 patients (cf. Figure 1B). However, reference #23 reports that "[i]n the escalation phase, 46 subjects received between 0.5–15.6 mg/kg of VT1021 by IV infusion twice weekly". Thus, it is not clear to me whether the 46 patients reported here are from the escalation phase, the expansion phase, or both, and at what dose they were treated.

>> This manuscript only covers the phase 1 expansion study. All patients received 11.8 mg/kg of VT1021 by IV infusion twice weekly (page 4). We apologize for the confusion and have revised the manuscript to better clarify (Title, Plan language summary and Abstract) (page 1).

A total of 47 evaluable subjects have been enrolled in this expansion study, one subject had day 1 samples damaged, therefore the baseline biomarker data is only available for 46 subjects. We have provided this information in the revised version (page 7).

5. In line 97, the authors state that "a pre-study or archival tumor specimen was collected". Can the authors confirm this sample was taken after the patient signed the consent form? The wording is somewhat ambiguous as archival could also relate to a sample taken at the time of initial diagnosis before any previous lines of treatment.

>> We confirm that no tumor biopsy sample, archived or pre-study was provided prior to the patient signing the informed consent. We have revised the manuscript to clarify this (Page 5).

6. The authors show boxplots with median values and outliers throughout the manuscript. However, there is no definition of what constitutes an outlier (which can be different from one statistical software package to another).

>> In the revised version, we have added a "Statistical analysis" section under the Methods to provide the detailed information for our statistical analyses (page 6). Additional information was added to each table/figure under the legend.

7. Regarding boxplots, the authors might want to consider a combination of violin and dot plots instead to allow the reader to assess entire distribution of a parameter. The tables indicating count, median and outliers could then be omitted with "count" being incorporated in the x-axis text (i.e., "CR/PR/SD (n=7)") and exceptionally interesting median values being stated in the main text.

>> We have revised Figure 1 to show the entire distribution of the fold changes by dot plots. We have replaced Figure 2 and 3 with Tables to show mean values and their confidence intervals for all groups.

8. ANOVA has been used for between-group comparisons. For some parameters, there are clear outliers which can bias the results of an ANOVA. Non-parametric approaches to statistical inference like the Mann-Whitney U test (for two groups) or the Kruskal-Wallis test (for more than two groups) would provide a more robust choice. This would also be in line with showing the median (a non-parametric measure) instead of the arithmetic mean. An alternative option would be to completely abstain from statistical significance testing for these exploratory analyses and give more emphasis to estimation and precision (that is, effect measures and their confidence intervals).

>> We thank the reviewer for the suggestions, we have revised the analysis by showing effect measures and their confidence intervals. We have replaced Figure 2 with Table 3, Figure 3 with Table 4.

9. In Figure 3, the authors present a total of 9 analyses highlighting the significant difference in secreted TSP-1 induction between CR/PR/SD and PD patients. Given that they do not seem to have corrected for multiple testing, this result may be due to chance and thus the conclusions drawn in lines 192-195 appear too strong.

>> We agree with the reviewer; we have revised the conclusions and indicated that: "... secreted TSP-1 levels... have potential correlation to the clinical response. It will be a specific focus to test this hypothesis with additional samples from future clinical studies." (Page 8).

10. In line 303, what do the authors mean with "more robust" fold changes? In general, it is not entirely clear according to what criteria these four biomarkers were selected from the 105 cytokines panel.

>> We have revised the manuscript to clarify that "more robust" means cytokines with higher fold changes than TSP-1 after VT1021 treatment. We have added more information to explain how the biomarkers were selected from the 105 cytokines panel (page 10).

11. In line 398, the authors state that "additional cytokines (...) showed a trend (...) to be biomarkers". Trends in the statistical sense can either be with regard to time or across the levels of an ordinal variable. Neither is the case here, so the term trend should be avoided.

>> We agree with the reviewer and have removed the "trend" (page 13).

12. The quality of the figures is suboptimal in general. Vertical axes should have the same scale if they are measuring the same (e.g., in Figure 1). Instead of showing asterisks to indicate p-values, the actual p-values should be shown. Figure 4C/4F don't seem to be the best way to represent these data (if a graphical representation is required at all).

>> We thank the reviewer for all the suggestions. We have revised Figure 1 to show dot plots and keep the same scale of vertical axes for all the graphs. We replaced Figure 2 and 3 with Tables to show effect measures and their confidence intervals. For Figure 4 (Figure 3 in the revised version), we have simplified the image presentation and data analysis (Page 8-9).

Reviewers' comments:

Reviewer #1 (Remarks to the Author):

Authors have responded to all the concerns which were raised during the initial review process. The manuscript is acceptable in its current form.

Reviewer #2 (Remarks to the Author):

The authors have correctly answered to my comments. The article is now ready for being accepted.

Reviewer #3 (Remarks to the Author):

The revised manuscript has greatly improved for clarity. I have a few comments for consideration.

- The response letter provided detailed explanations for the use of maximum or minimum analyte levels for fold change calculation and in-treatment comparison. It is recommended to be included as supplementary materials.

- Were all 7 biomarkers listed in Table 5 confirmed with ELISA assays? The message seems unclear: "In summary, four potential predicative biomarkers, MMP9, PAI-1, CHI3L1 and CCL5, were identified from a panel of 105 human cytokines and further confirmed with ELISA assays (Table 5). Five potential pharmacodynamic biomarkers, including PAI-1, MIF, IL18 Bpa, CHI3L1 and CCL5 (Table 5) were identified (Table 5)."

- Did biomarker analysis in TME show association of the change with VT1021 treatment? The message should be clarified.

Reviewer #4 (Remarks to the Author):

The authors have addressed most of my previous comments. However, some new issues have arisen:

- The newly introduced statistical methods section lacks the information of what constitutes a statistically significant result, i.e., what was the significance level for the statistical tests. Figure and table legends suggest it was set at 5%.
- The authors use one-sided statistical tests throughout the manuscript. It seems unlikely that this is justified in all cases, in other words that the group with the larger mean was known to be the group with the larger mean before the test was conducted. Two-sided tests should be used if there is even the smallest chance of any of the two groups having the larger mean. The authors present two-sided confidence interval, which contradicts their one-sided testing approach. If one-sided tests are used, the significance level should be set at half the two-sided significance level (here: 2.5%).
- In the newly introduced clinical response subsection (page 6/7), what are the confidence intervals for the respective disease control rates?
- The authors state that "TSP-1 induction by VT1021 has been observed for all cancer types tested here, and no significant difference was observed between different cancer types". However, the

statistical methods section lacks any tests that can compare more than two groups.

- Presenting only statistically significant p-values in Table 3 and other tables constitutes selective reporting; all p-values should be reported.
- There is no such thing as "p-values approaching statistical significance" and such wording should be avoided.
- In Table 3, no confidence intervals are given for TSP-1 ratios.
- Comparing subjects' maximum values versus healthy donor baseline values in Table 3 will be biased by random variation. Even if the group means are the same and stable over time, for the subjects always the highest value will be selected whereas for the healthy donors there is only one value available.
- Reviewer #3 had asked about the trend of the TSP-1 biomarker over time. The authors replied that there was no obvious trend in TSP-1 changes across the 15 timepoints, but that they observed significant TSP-1 changes after 50 days of treatment with VT1021. The analysis providing the latter conclusion, however, is deeply flawed. It is not clear what statistical test was used, but assuming it was a paired t-test as in the manuscript, this test comes with the assumption of independent observations. Clearly, measurements from the same patient over time are not independent from each other. The observed statistical significance plainly results from multiplying the available data and artificially increasing the sample size, achieved through pooling Day 1 & 4 and Day 53 measurements, respectively.
- Page 6: "Disease control rate control rate".
- Several instances: "Predicative" instead of "predictive".

Responses to reviewers' comments received on 11/3 on our manuscript (COMMSMED-23-0132B)

Reviewer #1 (Remarks to the Author):

Authors have responded to all the concerns which were raised during the initial review process. The manuscript is acceptable in its current form.

>> We appreciate the reviewer's time and effort in reviewing our response and are pleased to hear that we have addressed the comments. We thank the reviewer for their helpful suggestions throughout the review process.

Reviewer #2 (Remarks to the Author):

The authors have correctly answered to my comments. The article is now ready for being accepted.

>> We appreciate the reviewer's time and effort in reviewing our response and are pleased to hear that we have addressed the comments. We thank the reviewer for their helpful suggestions throughout the review process.

Reviewer #3 (Remarks to the Author):

The revised manuscript has greatly improved for clarity. I have a few comments for consideration.

- The response letter provided detailed explanations for the use of maximum or minimum analyte levels for fold change calculation and in-treatment comparison. It is recommended to be included as supplementary materials.

>> We thank the reviewer for the suggestion. We have added the detailed explanations as supplemental information (Supplemental Figure 1) (page 5).

- Were all 7 biomarkers listed in Table 5 confirmed with ELISA assays? The message seems unclear: "In summary, four potential predicative biomarkers, MMP9, PAI-1, CHI3L1 and CCL5, were identified from a panel of 105 human cytokines and further confirmed with ELISA assays (Table 5). Five potential pharmacodynamic biomarkers, including PAI-1, MIF, IL18 Bpa, CHI3L1 and CCL5 (Table 5) were identified (Table 5)."

>> Yes, all 7 biomarkers listed in Table 5 have been confirmed with ELISA assays. We have revised the manuscript to clarify this (page 11).

- Did biomarker analysis in TME show association of the change with VT1021 treatment? The

message should be clarified.

>> We have added a summary paragraph to discuss the biomarker analysis in the TME and its association with VT1021 treatment (page 10). With respect to the correlation of clinical response with changes to the TME for the GBM patient population, due to the location of the lesions, there is generally no possibility of obtaining in-treatment biopsy samples for the purpose of a clinical trial.

Reviewer #4 (Remarks to the Author):

The authors have addressed most of my previous comments. However, some new issues have arisen:

- The newly introduced statistical methods section lacks the information of what constitutes a statistically significant result, i.e., what was the significance level for the statistical tests. Figure and table legends suggest it was set at 5%.

>> Yes, the significance level for the statistical tests were set at 5%. We have revised the method section to include this information (page 6).

- The authors use one-sided statistical tests throughout the manuscript. It seems unlikely that this is justified in all cases, in other words that the group with the larger mean was known to be the group with the larger mean before the test was conducted. Two-sided tests should be used if there is even the smallest chance of any of the two groups having the larger mean. The authors present two-sided confidence interval, which contradicts their one-sided testing approach. If one-sided tests are used, the significance level should be set at half the two-sided significance level (here: 2.5%).

>> We thank the reviewer for the suggestions. We have revised the statistical analysis and used two-sided tests for Table 3, 4 and 5. Notice that after the revision of the statistical analysis, all major conclusions remain unchanged.

- In the newly introduced clinical response subsection (page 6/7), what are the confidence intervals for the respective disease control rates?

>> The reviewer's question is confusing: "what are the confidence intervals for the respective DCRs".

The DCR for GBM was 45%; for pancreatic cancer was 18%. It is impossible to calculate confidence intervals for disease control rates as they are simply the ratio of CR/PR/SD to total patients enrolled. There are no measurements involved that can generate imprecision with this calculation.

- The authors state that "TSP-1 induction by VT1021 has been observed for all cancer types tested here, and no significant difference was observed between different cancer types". However, the statistical methods section lacks any tests that can compare more than two groups.

>> We have revised the method section to include that we used one-way analysis of variance (ANOVA) for Figure 2 (page 6).

- Presenting only statistically significant p-values in Table 3 and other tables constitutes selective reporting; all p-values should be reported.

>> We have revised Table 3-5 and reported all p-values.

- There is no such thing as "p-values approaching statistical significance" and such wording should be avoided.

>> We have revised the manuscript and removed "approaching statistical significance".

- In Table 3, no confidence intervals are given for TSP-1 ratios.

>> We have revised Table 3 to report confidence intervals for TSP-1 ratios.

- Comparing subjects' maximum values versus healthy donor baseline values in Table 3 will be biased by random variation. Even if the group means are the same and stable over time, for the subjects always the highest value will be selected whereas for the healthy donors there is only one value available.

>> This statement is simply incorrect. We are not introducing bias. There are no reports indicating that there are time dependent fluctuations in the TSP-1 levels in healthy adults, therefore the healthy donor has a steady state level of TSP-1. If we were measuring blood glucose, which does vary with time of day or food intake, that would be a different matter. The group means are not the same and stable over time, in fact we show that the induction is time dependent and increases following administration of VT1021. Accordingly, we used the maximal induction level of TSP-1 for patients as this is the only way to take into account the biological activity of VT1021. As this was an oncology clinical trial, a healthy volunteer population was not enrolled as part of the study.

- Reviewer #3 had asked about the trend of the TSP-1 biomarker over time. The authors replied that there was no obvious trend in TSP-1 changes across the 15 timepoints, but that they observed significant TSP-1 changes after 50 days of treatment with VT1021. The analysis providing the latter conclusion, however, is deeply flawed. It is not clear what statistical test was used, but assuming it was a paired t-test as in the manuscript, this test comes with the assumption of independent observations. Clearly, measurements from the same patient over

time are not independent from each other. The observed statistical significance plainly results from multiplying the available data and artificially increasing the sample size, achieved through pooling Day 1 & 4 and Day 53 measurements, respectively.

>> The purpose of showing all datapoints across all time points was to understand the biomarker change(s) over time and explain how we decided when to use Max value or Min value. The purpose wasn't to emphasize statistical difference. Therefore, we have removed the p-value calculation in the revised version (Supplemental Figure 1).

- Page 6: "Disease control rate control rate".

>> Thanks; we have revised the manuscript (page 6).

- Several instances: "Predicative" instead of "predictive".

>> Thanks; we have revised the manuscript (page 11, 13).

Reviewers' comments:

Reviewer #3 (Remarks to the Author):

Only one minor comment: Please define TSP-1 fold change in Table 4.

Reviewer #4 (Remarks to the Author):

The authors have addressed most of my comments. I am happy to provide below some clarifications of outstanding comments.

1. The authors argue that it "is impossible to calculate confidence intervals for disease control rates as they are simply the ratio of CR/PR/SD to total patients enrolled." This definition of disease control rates defines a binomial proportion. Similar to other proportions like mortality within X years or overall response rate during the trial, confidence intervals can of course be calculated for disease control rates (https://en.wikipedia.org/wiki/Binomial_proportion_confidence_interval). There are several approaches, most of them are very simple and many are implemented in statistical software or available in online calculators.

2. The bias introduced by the maximum approached used for the analyses in Table 3 is less associated with the healthy donor group rather than the within-patient variation. The authors' analysis is indeed unbiased under the assumption that there is no within-patient variation. Such variation can result from physiological variation of any degree (not necessarily related to an upward/downward trend over time), measurement error, inter-machine variability, inter-rater variability, or other sources. In other words, if a patient has a steady-state level of TSP-1, the readouts of each timepoint will have to be exactly the same for a patient. A scenario that is unlikely in most cases. Where there is a trend over time in patient's TSP-1 levels, what is random variation and what is not becomes more difficult to assess. If such random variation exists, however, the maximum value chosen from a range of timepoints is more likely to be an outlier. Any statistician would be able to draw up an illustrating example in very little time. The approach is not a major flaw, but it does have the limitation of potentially introducing bias.

Responses to reviewers' comments received on 12/15 on our manuscript (COMMSMED-23-0132C)

Reviewers' comments:

Reviewer #3 (Remarks to the Author):

Only one minor comment: Please define TSP-1 fold change in Table 4.

>> We have added the definition of TSP-1 fold change in Table 4.

Reviewer #4 (Remarks to the Author):

The authors have addressed most of my comments. I am happy to provide below some clarifications of outstanding comments.

1. The authors argue that it "is impossible to calculate confidence intervals for disease control rates as they are simply the ratio of CR/PR/SD to total patients enrolled." This definition of disease control rates defines a binomial proportion. Similar to other proportions like mortality within X years or overall response rate during the trial, confidence intervals can of course be calculated for disease control rates (https://en.wikipedia.org/wiki/Binomial_proportion_confidence_interval). There are several approaches, most of them are very simple and many are implemented in statistical software or available in online calculators.

>> We thank the reviewer for the suggestion and explanation. We have calculated the confidence intervals of DCRs using Clopper-Pearson Exact method and added the information in the revised manuscript (page 2, 3, 6 &7)

2. The bias introduced by the maximum approached used for the analyses in Table 3 is less associated with the healthy donor group rather than the within-patient variation. The authors' analysis is indeed unbiased under the assumption that there is no within-patient variation. Such variation can result from physiological variation of any degree (not necessarily related to an upward/downward trend over time), measurement error, inter-machine variability, inter-rater variability, or other sources. In other words, if a patient has a steady-state level of TSP-1, the readouts of each timepoint will have to be exactly the same for a patient. A scenario that is unlikely in most cases. Where there is a trend over time in patient's TSP-1 levels, what is random variation and what is not becomes more difficult to assess. If such random variation exists, however, the maximum value chosen from a range of timepoints is more likely to be an outlier. Any statistician would be able to draw up an illustrating example in very little time. The approach is not a major flaw, but it does have the limitation of potentially introducing bias.

>> We understand the limitation of the analysis using Max value. We have discussed this limitation in the revised manuscript (page 5).